# Resolving Extreme Data Scarcity by Explicit Physics Integration: An Application to Groundwater Heat Transport

## Abstract

Machine learning methods often struggle with real-world applications in science and engineering due to an insufficient amount or quality of training data. In this work, the example of subsurface porous media flow is considered; this corresponds to advection-diffusion processes under heterogeneous flow conditions, i.e., for spatially varying material parameters, and a large number of spatially distributed source terms. This challenge comes at high computing costs for classical simulation methods due to the required high spatio-temporal resolution and large domains. Machine learning-based surrogate models seem to offer a computationally efficient alternative. However, faced with real-world data-limitations, purely data-driven approaches face difficulties in predicting the advection process, which is highly sensitive to input variations and involves long-range interactions. Therefore, in this work, a Local-Global Convolutional Neural Network (LGCNN) approach is introduced, that combines a lightweight numerical surrogate for the global transport process with convolutional neural networks (CNNs) for the local processes. With the LGCNN, we model a city-wide subsurface temperature field, involving a heterogeneous groundwater flow field and one hundred groundwater heat pump injection points forming interacting heat plumes over long distances. In order to first systematically analyze the method, random subsurface input fields are employed. Then, the model is trained on a few cut-outs from a real-world subsurface map of the Munich region in Germany. Our model scales to larger cut-outs without retraining by accounting for the global effects with numerical physics models. All datasets, our code, and trained models are published for reproducibility.

## 1 Introduction

Many real-world systems, across environmental engineering, geoscience, and biomedicine, feature transport phenomena governed by coupled advection–diffusion processes, e.g., pollutant transport through air or water, chemical reactions, or heat transport in flowing media. In such settings, interactions can span large distances, long timescales and many components.

Datasets based on physical measurements are often limited by cost, logistics, or experimental constraints. While numerical simulations can address these limitations to some extent, they become impractically slow or computationally expensive as domain size or resolution increases. Machine learning (ML) models offer fast predictions but typically require large training datasets and often struggle to scale to larger domains. This trade-off underscores the need for data-efficient surrogates that can capture both local and long-range interactions, and scale to arbitrary domain sizes and numbers of interacting components. To this end, we introduce the **Local–Global Convolutional Neural Network (LGCNN)**, a scalable, physics-inspired approach that leverages the structure of advection–diffusion problems by decoupling local and global transport processes.

We apply this approach to the climate change-relevant application of modeling city-wide subsurface temperature fields driven by heat and mass injections of dozens of open-loop groundwater heat pumps (GWHP) (Pophillat et al., 2020; Department-of Energy, 2023; Hähnlein et al., 2013) in the region of Munich, Germany (Zosseder et al., 2022). A city-wide optimization of the positions of all GWHPs requires calling a prediction model dozens of times with slightly adapted GWHP positions to minimize negative interactions between different heat plumes and pumps. For this scenario, our model should predict the temperature field $T$ in an arbitrarily large domain based on the inputs of the subsurface parameters of a hydraulic pressure gradient $\nabla p$ and a heterogeneous subsurface permeability field $k$, and arbitrary heat pump locations $i$. Labels are generated by expensive simulations, but only for a small number of cases and entirely offline.

As sketched in Figure 1, the LGCNN uses CNNs where they excel, and simple numerical surrogates where CNNs struggle: a first CNN predicts the velocities $\vec{v}$ from local relations of the inputs $p$, $k$ and $i$ (Step 1); a fast numerical component calculates the streamlines $\vec{s}$ based on this (Step 2); a second CNN calculates the temperature distribution $T$ in the whole domain based on all prior inputs and outputs (Step 3). Since Step 2 covers all non-local effects, Step 3 can again take advantage of CNNs to learn local patterns. We outline related work in Section 2, then define our

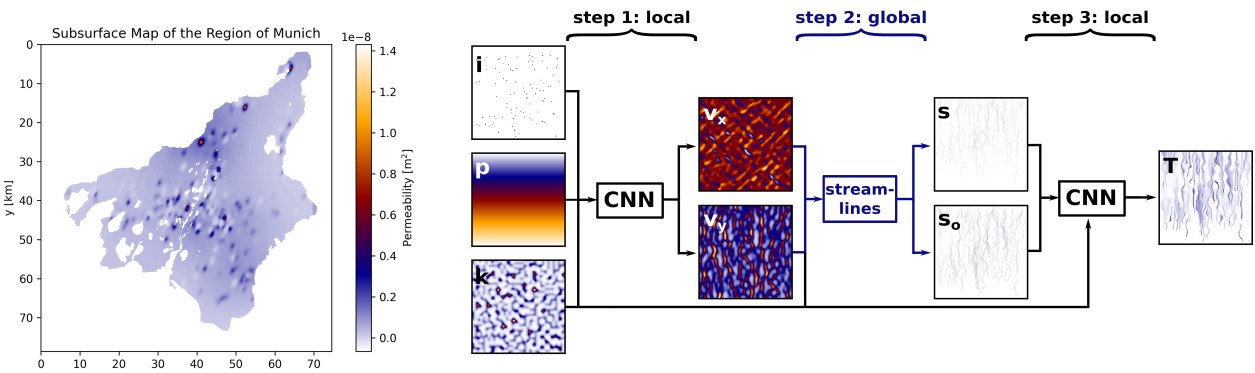

Figure 1: Left: Map of $k$ of whole region of Munich. Right: Schematics of our Local-Global CNN-based approach (LGCNN) with 3 physics-inspired steps: CNN ($pki \rightarrow \vec{v}$), simplified solver ($i\vec{v} \rightarrow \vec{s}$), CNN ($pki\vec{v}\vec{s} \rightarrow T$).

datasets and metrics in Section 3. In Section 4, we benchmark purely data-driven models and show their performance and limitations on this low-data challenge. To address data scarcity, we introduce LGCNN in Section 5. Section 6 emphasizes generalization and scalability from a single datapoint, while Section 7 demonstrates transfer to real-world subsurface parameters extracted from measurement maps such as in Figure 1.

**Contributions**  Our method strongly reduces data requirements for modeling complex scenarios, increases reliability, and ensures scalability to larger domains by a domain-specific modular design of physics-aided machine learning for an input-sensitive scenario with long-distance effects. Subsurface temperature prediction serves as our example application, with the approach expected to generalize to related domains.

**Limitations**  The approach is best suited for systems where temperature-induced flow changes are small or spatially confined. It is currently restricted to two-dimensional and steady-state datasets. While the underlying methodology readily generalizes to three-dimensional and transient scenarios, such extensions are not yet implemented due to the absence of appropriate training data.

## 2    RELATED WORK

Deep learning surrogates in scientific modeling can be broadly categorized into purely data-driven architectures, physics-informed or domain-structured models, and highly problem-specific hybrid approaches. In this section, we summarize the developments in each category and existing efforts in GWHP modeling.

Convolutional Neural Networks (CNNs), particularly UNet (Ronneberger et al., 2015a) variants, are widely used for spatial prediction tasks due to their locality-aware architecture of moving-window-like kernels. Applications include flow around airfoils (Thuerey et al., 2020) and various engineering systems (Jhaveri et al., 2022). For general reviews on ML in scientific and real-world engineering contexts, see Sharma et al. (2021); Sarker (2021); Angra & Ahuja (2017).

To improve generalization and reduce data requirements, domain-structured models embed physical priors or symmetries into their architecture. This includes Fourier Neural Operators (FNOs) for efficient multi-scale modeling (Li et al., 2020; Choi et al., 2024), thermodynamics-preserving networks (Hernández et al., 2021), and rotation-equivariant CNNs (ECNNs) (Weiler et al., 2023). Physics-Informed Neural Networks (PINNs) go further by replacing data loss with physics-based constraints, allowing for training without labeled data (Raissi et al., 2019; Cuomo et al., 2022), and have been applied to fluid dynamics and inverse problems (Rao et al., 2020; Sun et al., 2020; Cai et al., 2021).

Hybrid approaches combine architectures and integrate domain or physics knowledge to improve general models, for instance, in low-data regimes. Examples include staged CNNs for atmospheric plume dispersion (Fernández-Godino et al., 2024), physics-informed FNOs for traffic flow (Thodi et al., 2024), CNN–MLP hybrids for optimizing positions of oil wells (Yousefzadeh et al., 2025), and physics-guided CNNs for seismic response prediction (Zhang et al., 2020).

**Deep Learning for GWHP Modeling**  In the context of groundwater modeling with heat transport, recent work focuses on isolated or pairwise interacting GWHPs. Most use UNet-based architectures, optionally with physics-loss terms (Davis et al., 2023; Pelzer & Schulte, 2024; Scheurer, 2021). Although effective in simple, homogeneous aquifers, these models rely on large training datasets or simulated inputs and have not been scaled to city-wide domains with many pumps. UNets are the dominant architecture because of their strong performance on spatial data, making them the baseline for comparison.

We briefly evaluated PINNs and FNOs as alternatives. PINNs, while promising for low-data fluid dynamics tasks (Rao et al., 2020; Sun et al., 2020; Cai et al., 2021; Takamoto et al., 2022), struggled with complex scenarios (cf. Krishnapriyan et al. (2021)) such as heterogeneous media and discontinuous source terms in preliminary tests. Specifically, we tested PINNs in three simplified single-plume settings: on a homogeneous 2D aquifer, a heterogeneous 2D aquifer, and a homogeneous 3D aquifer. In all cases, PINNs failed to outperform standard CNNs, even when the physics loss was only used as a regularization term. The most pronounced discrepancies occurred near injection points, likely due to discontinuities in the governing equations. FNOs, though capable of modeling local and global dependencies, struggled with large domain sizes due to high memory requirements. Additionally, they performed poorly in the presence of multiple sources, consistent with Liu-Schiaffini et al. (2024).

# 3 DATASETS AND METRICS

**Datasets**  Inputs for the neural networks consist of a heterogeneous permeability field $k$, an initial hydraulic pressure field $p$, and a one-hot-encoded field of heat pump positions $i$. The (interim) labels of velocity $\vec{v}$ and temperature $T$ fields are simulated, in our case with Pflotran (Lichtner et al., 2015a), until a quasi-steady state is reached after $\approx 27.5$ years simulated time (Umweltministerium Baden-Württemberg, 2009). All data are normalized to $[0, 1]$, and stored in PyTorch format for training.

We generate two types of datasets, one is based on synthetic, the other on real permeability fields $k$. Both cover a $12.8 \times 12.8$ km$^2$ domain with $2\,560 \times 2\,560$ cells. The **baseline dataset** uses Perlin noise (Perlin, 1985) to generate random, heterogeneous $k$. Three simulations (*3dp*) with different fields for $i$ and $k$ are run, generating one datapoint each for training, validation, and testing, plus a simulation $4\times$ larger to assess scalability. For training vanilla neural networks in Section 4, we generate an additional dataset of 101 datapoints (*101dp*), split into 73:18:10 for train:val:test. Runtimes are $\approx 27$ hours per simulation and 123 hours for the larger domain.

The more **realistic dataset** builds on $k$ fields cut from maps of borehole measurements in the Munich region (Bayerisches Landesamt für Umwelt, 2015). This dataset consists of four simulations (three for training and one for validation) and one larger simulation for testing scalability. Due to constraints of the available measurements, see Figure 1, the large-scale simulation extends only in length, resulting in a rectangular domain twice the length. Runtimes range from 38 to 91 hours (average 58 hours), and 134 hours for the scaling-test domain. Variation across the dataset stems from different heat pump placements in $i$ and from the specific regions where $k$ is extract from the available measurement domain.

For detailed information on simulation procedures and hydro-geological parameters, see Appendix A.1; for hardware specifications see Appendix A.4.

**Performance Metrics**  We evaluate model accuracy per output dimension $(v_x, v_y, T)$ separately using Mean Absolute Error (MAE) (Naser & Alavi, 2023), Mean Squared Error (MSE) (Naser & Alavi, 2023), Maximum Absolute Error ($L_\infty$) Structural Similarity Index Measure (SSIM) (Wang et al., 2004), and application-driven metrics of Percentage Above Threshold (PAT) and Visual Assessment. PAT measures the percentage of cells where the absolute error of the predicted temperature exceeds the threshold of $0.1$ °C, corresponding to measurement precision (UKB System Technology), and is only applicable to temperature predictions. All metrics other than SSIM are applied after re-normalization to the original data ranges to obtain physically meaningful results in [°C] or [meters/year].

## 4    APPLICATION OF PURELY DATA-DRIVEN APPROACHES

In this section, we present the performance of two purely data-driven approaches for predicting $T$ from inputs of $p, k, i$: a UNet (Ronneberger et al., 2015a) and the domain-decomposition-based UNet variant, called DDUNet (Verburg et al., 2025).

**Methods**    The fully convolutional vanilla UNet has an encoder-decoder architecture with skip connections between layers of equal spatial dimensions. Convolutional networks excel in extracting local dependencies from spatial data with locally repeating patterns. Hence, in theory, one datapoint covering a large domain could be enough to generalize, if only local dependencies occur. Global patterns would require a large receptive field and, hence, more data. The DDUNet is designed to handle large-scale domains under GPU memory constraints. It combines a UNet architecture with domain decomposition principles by partitioning the spatial domain into subdomains that are distributed across multiple GPUs. We use a decomposition into $2 \times 2$ subdomains.[1] Global context is communicated between subdomains via coarse feature maps through a learned communication network, maintaining spatial consistency and providing a large receptive field while reducing memory demands. The hyperparameters of the models are optimized according to Appendix A.2.1.

**Experiments**    Experiments for the vanilla UNet are conducted on the baseline datasets with *3dp* and *101dp*, those with DDUNet directly on the *101dp* dataset, see Appendix A.3. Both models are applied to the test of *3dp* and to the scaling datapoint, for direct comparability to LGCNN. UNet exhibits poor performance when trained on the limited *3dp* dataset (UNet$_{3dp}$), as demonstrated in both Figure 4 and Table 1. Similar performance would be expected for the DDUNet when applied to this sparse dataset. When trained on *101dp*, however, both demonstrate robust predictive and scaling capabilities. The advantage of DDUNet over vanilla UNet is a reduction of memory usage and runtime, see appendix A.3.

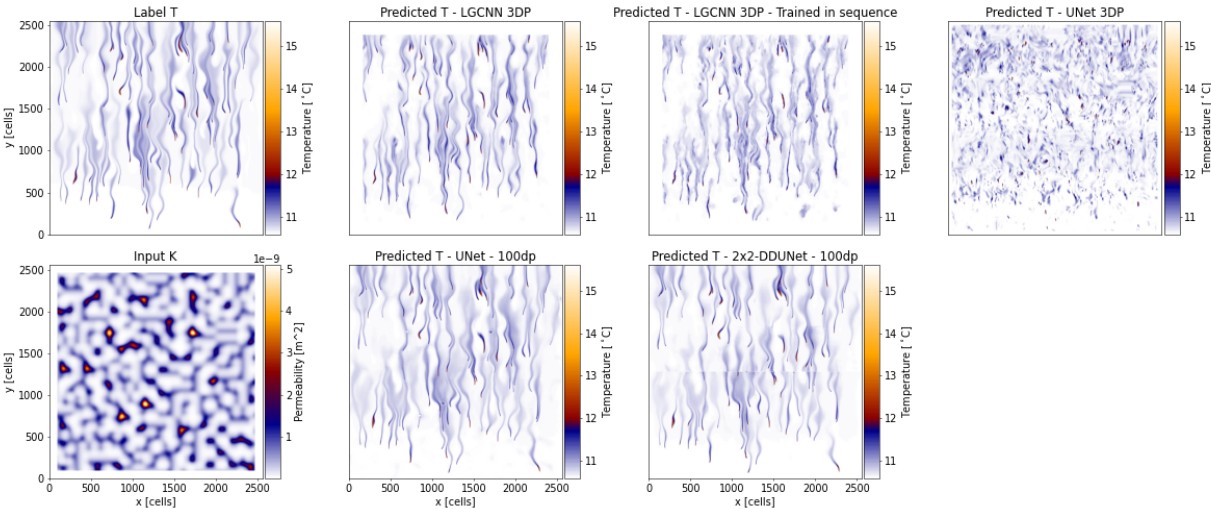

Figure 2: $pki \rightarrow T$ (baseline dataset (*3dp*), test). 1st column: Label $T$ and input $k$. 2nd-4th column: LGCNN$_{3dp}$ (isolated Step 3 and full pipeline, see Sections 5 and 6), vanilla UNet and DDUNet on *3dp* and *101dp* (Section 4).

## 5    METHODOLOGY

In data-scarce settings, performance can improve when the structure of a model reflects the underlying physics, cf. Section 2. This section hence outlines the relevant physical processes, motivates our LGCNN method, and discusses a few details of LGCNN's individual components.

---

[1]Conceptually, DDUNet reduces to a standard UNet when trained with a single subdomain.

Table 1: Performance metrics for predicting $T$ from $pki$ with UNet and DDUNet. Errors in [°C], MSE in [°C$^2$], PAT in [%], and SSIM unitless. Inferred on the test of *3dp* and the scaling datapoint.

| Model | Test | | | | | Scaling | | | | |
|---|---|---|---|---|---|---|---|---|---|---|
| | $L_\infty$ | MAE | MSE | PAT | SSIM | $L_\infty$ | MAE | MSE | PAT | SSIM |
| UNet$_{3dp}$ | 4.8642 | 0.1314 | 0.0492 | 39.05 | 0.5794 | – | – | – | – | – |
| UNet$_{101dp}$ | 4.3985 | 0.0473 | 0.0100 | 13.63 | 0.9827 | 4.3426 | 0.0202 | 0.0033 | 4.33 | 0.9955 |
| $2 \times 2$-DDUNet$_{101dp}$ | 3.4257 | 0.0548 | 0.0128 | 17.42 | 0.9804 | 4.1806 | 0.0235 | 0.0052 | 6.11 | 0.9940 |

**Physics of Groundwater Flow with Heat Transport**   Transport of heat induced by heat injections of GWHPs in the subsurface is an advection–diffusion process. Since the Péclet number, i.e., the advection-diffusion ratio, is high in our application (Pe≫1, more in Appendix A.1), the system is advection-dominated. Advection is largely governed by the global hydraulic pressure gradient, driving flow from higher to lower regions. Locally, the flow paths are influenced by the spatial distribution of permeability $k$, cf. Figure 2. As a result, small changes in $k$ can lead to significant differences in flow paths further downstream, demonstrating high input sensitivity.

**One-Way Coupled Approach: Local-Global CNN (LGCNN)**   In a fully coupled physical model, flow field computation and heat transport along streamlines (NASA) starting at heat pumps (advection) and their widening (diffusion) must be solved monolithically. By splitting effects into local and global and working with steady-state simulated $\vec{v}$ during training, we can reduce the coupling to one direction without significant loss in accuracy. Our simplified physical pipeline consists of three steps: (1) compute a steady-state flow field $\vec{v}$ from initial subsurface parameters $p$ and $k$, with $i$ encoding mass influx around GWHPs; (2) transport injected heat along streamlines governed by $\vec{v}$ until quasi-steady state (Umweltministerium Baden-Württemberg, 2009); and (3) apply plume widening to these heat paths, informed by soil diffusivity and $\vec{v}$, to approximate diffusion effects. The resulting one-way coupled LGCNN approach (see Figure 1) can formally be described as:

Step 1 **Velocities (local)** We employ a CNN to predict the velocity field $\vec{v} = (v_x, v_y)$ from $p$, $k$, and $i$:

$$\text{CNN}\,(p, k, i) = \vec{v}. \tag{1}$$

Step 2 **Streamlines (global)** Based on the predicted velocities $\vec{v}$ from Step 1, we compute streamlines $\vec{s}$ originating from all pump locations in $i$ with an initial value problem (IVP) solver:

$$\text{IVP}\,(i, \vec{v}) = \vec{s}. \tag{2}$$

Step 3 **Temperature (local)** A second CNN predicts the temperature field $T$ from the initial inputs $p$, $k$, $i$, the predicted velocities $\vec{v}$ of Step 1 and streamlines $\vec{s}$ calculated in Step 2:

$$\text{CNN}\,(p, k, i, \vec{v}, \vec{s}) = T. \tag{3}$$

The model outputs the steady-state temperature field $T$ and, using the same inputs $p$, $k$, and $i$ as a simulation, serves as its direct surrogate.

**CNN Models in the Local Steps**   The local Steps 1 and 3 are approximated by a UNet (Ronneberger et al., 2015a; Pelzer & Schulte, 2024), see Appendix A.2 for details of our architecture. It is beneficial to omit zero-padding to enforce shift invariance and reliance on purely local information; this approach is also validated by Islam et al. (2020). Due to the input sensitivity of the entire problem, each step is trained independently on simulated input data. In particular, the CNN in Step 3 is trained on the simulated velocities $\vec{v}_{\text{sim}}$ and streamlines based on these fields. Only during inference, all steps are applied sequentially, i.e., Step 3 uses the outputs of Steps 1 and 2, including the predicted $\vec{v}_{\text{pred}}$ as inputs.

The training is accelerated by partitioning the data into overlapping patches, which increases the effective number of datapoints while reducing their spatial size. Combined with the streamlines' embedding and localizing of global flow patterns for the fully convolutional (locally) acting CNN, this training data enrichment allows the model to train effectively on very little data, sometimes as little as a single simulation run, while generalizing to unseen and even to larger domains. Validation and testing are conducted on independent simulation runs, with the full domain being processed as a whole instead of patches during inference.

The architecture, input choices, and training hyperparameters — including the patch size and overlap in the datapoint partitioning — were optimized through a combination of manual and automated tree-based search using Optuna (Akiba et al., 2019); see Appendix A.2.1 for details.

**Streamline Calculation and 2D-Embedding in the Global Step**   Streamlines are calculated by solving the initial value problem (IVP)

$$\frac{dy}{dt} = \vec{v}(y), \text{ with } y(t_0) = y_0, \tag{4}$$

with a lightweight numerical solver for each heat pump, where $y_0$ represents the location of a heat pump in $i$. We employ the `solve_ivp` function from SciPy (sci, a) with an implicit fifth-order Runge-Kutta method (sci, b) and 10 000 time steps of one day, corresponding to the simulation time of forming a heat plume.

The computed streamlines $y$ represent a sequence of positions in 2D, that a particle - injected at a heat pump's location - would traverse based on $\vec{v}$. The corresponding grid cells are assigned values that fade linearly from 1 to 0, reflecting the time it takes to reach each position, similar to a soft one-hot encoding.

To address the sensitivity of this process with respect to $k$ and $\vec{v}$, we calculate a set of outer streamlines $s_o$. For this, we perturb each heat pump's location $y_0$ by 10 cells orthogonal to the global flow direction, i.e., $\nabla p$, and compute the corresponding streamlines. By incorporating $\vec{s} = (s, s_o)$, the subsequent Step 3 obtains a spatial, localized representation of flow paths. In future work, we aim to explore probabilistic perturbations to compute the mean and standard deviation of streamlines, which, while being computationally more expensive, could be efficiently parallelized on GPUs.

## 6   RESULTS ON SYNTHETIC PERMEABILITY FIELDS

This section evaluates our model's performance on the baseline dataset *3dp* and on a scaling test datapoint; cf. Section 3. In Section 6.1, we present metrics for isolated velocity (Step 1) and temperature (Step 3) predictions, and for the full pipeline, demonstrating the potential of our approach. In ablation studies (Section 6.2), we motivate model design choices. Additional metrics and visualizations can be found in Appendix A.3.

Table 2: Metrics for predicting $\vec{v}_{\text{pred}}$ in Step 1 and in ablation studies of Step 1 on test and on scaling data from our baseline *3dp* dataset with random $k$. In [m/y], except for MSE in [m$^2$/y$^2$], SSIM unitless.

| | Output | Case | $L_\infty$ | MAE | MSE | SSIM |
|---|---|---|---|---|---|---|
| | **Step 1** | | | | | |
| LGCNN | $v_x$ | test | 190.8046 | 22.3178 | 972.5668 | 0.9911 |
| | $v_y$ | test | 256.2519 | 32.7444 | 2031.3357 | 0.9812 |
| | $v_x$ | scaling | 294.0457 | 24.9261 | 1204.1154 | 0.9911 |
| | $v_y$ | scaling | 367.6891 | 26.2795 | 1463.8218 | 0.9820 |
| | Train without data partitioning | | | | | |
| | $v_x$ | test | 202.6397 | 23.8584 | 1144.4479 | 0.9913 |
| | $v_y$ | test | 332.9557 | 35.9249 | 2490.4104 | 0.9789 |
| | Replace Step 1 with UNet$_{101dp}$ | | | | | |
| Ablation study | $v_x$ | test | 1047.8464 | 12.7036 | 896.2758 | 0.9016 |
| | $v_y$ | test | 2594.9504 | 14.3878 | 2317.6863 | 0.9903 |
| | $v_x$ | scaling | 659.8439 | 5.2152 | 79.2508 | 0.9699 |
| | $v_y$ | scaling | 456.1904 | 5.7492 | 83.9735 | 0.9948 |
| | Replace Step 1 with 2 × 2-DDUNet$_{101dp}$ | | | | | |
| | $v_x$ | test | 2453.5881 | 13.6735 | 1367.9061 | 0.9325 |
| | $v_y$ | test | 2029.0269 | 15.1332 | 2221.4226 | 0.9919 |
| | $v_x$ | scaling | 421.4186 | 8.1184 | 125.9511 | 0.9314 |
| | $v_y$ | scaling | 400.2490 | 7.4200 | 135.7929 | 0.9955 |

Table 3: Metrics for predicting $T$ in Step 3, in the full pipeline, and in ablations studies of Step 3 on test and on scaling data from our baseline *3dp* dataset with random $k$. In [°C], except for MSE in [°C$^2$], PAT in [%], SSIM unitless.

| | Case | $L_\infty$ | MAE | MSE | PAT | SSIM |
|---|---|---|---|---|---|---|
| | **Step 3**, i.e., test on $\vec{v}_{\text{sim}}$ | | | | | |
| LGCNN | test | 2.8990 | 0.0347 | 0.0041 | 7.54 | 0.9304 |
| | scaling | 3.0250 | 0.0168 | 0.0014 | 2.05 | 0.9510 |
| | **Pipeline**, i.e., test on $\vec{v}_{\text{pred}}$ | | | | | |
| | test | 4.2120 | 0.0905 | 0.0307 | 28.92 | 0.7637 |
| | scaling | 4.9366 | 0.0413 | 0.0141 | 10.87 | 0.8654 |
| | Train in sequence | | | | | |
| | test | 4.0804 | 0.0901 | 0.0289 | 29.44 | 0.7553 |
| | Include zero-padding | | | | | |
| | test | 2.3391 | 0.0418 | 0.0048 | 9.19 | 0.8838 |
| Ablation study | Train without data partitioning | | | | | |
| | test | 3.6420 | 0.0351 | 0.0043 | 7.41 | 0.9245 |
| | Replace Step 3 with UNet$_{101dp}$ | | | | | |
| | test | 4.4658 | 0.0416 | 0.0067 | 11.69 | 0.9835 |
| | scaling | 4.3489 | 0.0176 | 0.0022 | 3.07 | 0.9954 |
| | Replace Step 3 with 2 × 2-DDUNet$_{101dp}$ | | | | | |
| | test | 4.2772 | 0.0376 | 0.0058 | 10.96 | 0.9915 |
| | scaling | 3.9946 | 0.0192 | 0.0025 | 4.73 | 0.9968 |

### 6.1   PERFORMANCE OF LGCNN

**Isolated Steps 1 and 3**   Performance metrics for Step 1 are shown in Table 2, for Step 3 (trained and tested on simulated $\vec{v}_{\text{sim}}$) in Table 3. The combination of quantitative (MAE of (22.3, 32.7) m/y and 0.03 °C) and qualitative results through a visual assessment of Figure 3 demonstrate a strong performance with heat plumes forming according to $k$.

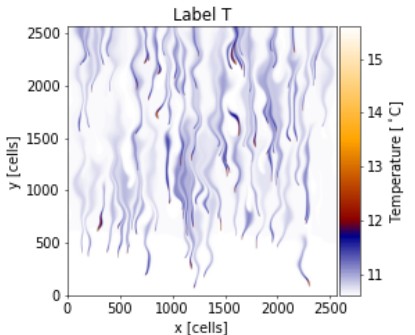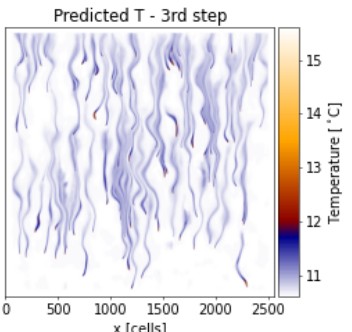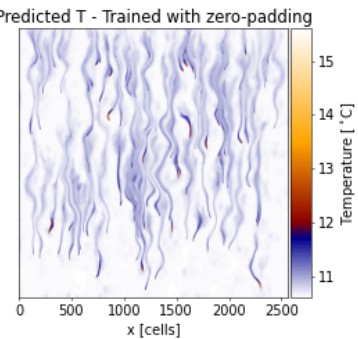

Figure 3: $pki\vec{s}\vec{v}_{\text{sim}} \to T$ (baseline dataset (*3dp*), test). 1st column: Label $T$. 2nd-3rd column: LGCNN Step 3, trained without and with zero-padding.

**Full Pipeline** Testing the full pipeline, i.e., Steps 2 and 3 are based on $\vec{v}_{\text{pred}}$, results in a higher test MAE of 0.09 °C compared to an isolated Step 3 (see Table 3). This was expected as the isolated Step 3 uses the true, simulated velocity fields $\vec{v}_{\text{sim}}$ as inputs. Visual assessment of representative predictions in Figure 2 reveals physically plausible heat plumes in terms of shape, extent, and heat magnitude. Deviations in streamlines arise from smaller errors in the velocity predictions from Step 1 $\vec{v}_{\text{pred}}$, highlighting the input sensitivity, especially near bifurcations, where small perturbations in $\vec{v}$ can lead to an alternative path around a clay lens in $k$. Compared to fully data-driven models trained on *101dp* (see Table 1), our model achieves a similar accuracy while training on only 1 versus 73 training datapoints, which strongly reduces computation time (see Table 16 for details).

**Scaling Test** We can already expect that our model is able to scale to larger domains, as we train only on local patches of the spatial domain and achieve high test accuracies on the full domain during validation and testing; see also Section 6.2 *Partitioning Training Data* for more details. To further demonstrate the scaling, we test our model on a domain of $4\times$ the size of the training domain with the same number of heat pumps, hence with a lower density of GWHPs/km$^2$; cf. Figure 4. We obtain a comparable MAE of 0.02 °C for Step 3, and an MAE of 0.04 °C for the full pipeline, see Table 3. A visual assessment of Figure 4 yields similar qualitative behavior as the previous results in Figure 2.

### 6.2 ABLATION STUDY

**Training in Sequence** For training Step 3, we can either employ simulated $\vec{v}_{\text{sim}}$ or predicted velocities $\vec{v}_{\text{pred}}$ as inputs. As illustrated in Figure 2, training on $\vec{v}_{\text{pred}}$ leads to physically implausible temperature fields with noisy artifacts and fragmented plumes. This likely stems from local misalignments between the streamline inputs $\vec{s}$ and the corresponding temperature label $T$, which strongly complicates localized training. Although loss values in Table 3 appear similar, the resulting temperature fields are less physically consistent and resemble those produced by end-to-end inference of UNet$_{3dp}$ in Section 4. In contrast, training with simulated velocities $\vec{v}_{\text{sim}}$ yields streamlines consistent with temperature fields, providing informative gradients and enabling stable learning.

**Zero-Padding** Removing zero-padding from all convolutional layers yields smaller but cleaner output fields, as shown in Figure 3. As a result, we obtain improved performance, reflected in lower PAT and higher SSIM scores in Table 3.

**Partitioning Training Data** Hyperparameter search shows the best performance for partitioning the domain during training into 20 736 overlapping patches of 256×256 cells for Step 1 and 82 944 patches for Step 3. Although this increases per-epoch compute time compared to full-domain training, it significantly reduces the number of epochs needed to reach comparable test performance (cf. Tables 2 and 3), resulting in an overall training time reduction of 67–90% (cf. Table 15).

**Replacing Isolated Steps with UNet$_{101dp}$ and $2 \times 2$-DDUNet$_{101dp}$** We individually replace each of the three steps with an optimized (cf. Appendix A.2.1) vanilla approach, i.e., UNet or DDUNet trained on *101dp*. For Step 2, both fail completely; results for Steps 1 and 3 are summarized in Tables 2 and 3. In Step 1, UNet slightly outperforms DDUNet, while the reverse holds for Step 3. A visual comparison of the scaling datapoint in Figure 4 reveals that increasing training data from *3dp* to *101dp* has minimal impact on the performance in Step 3.

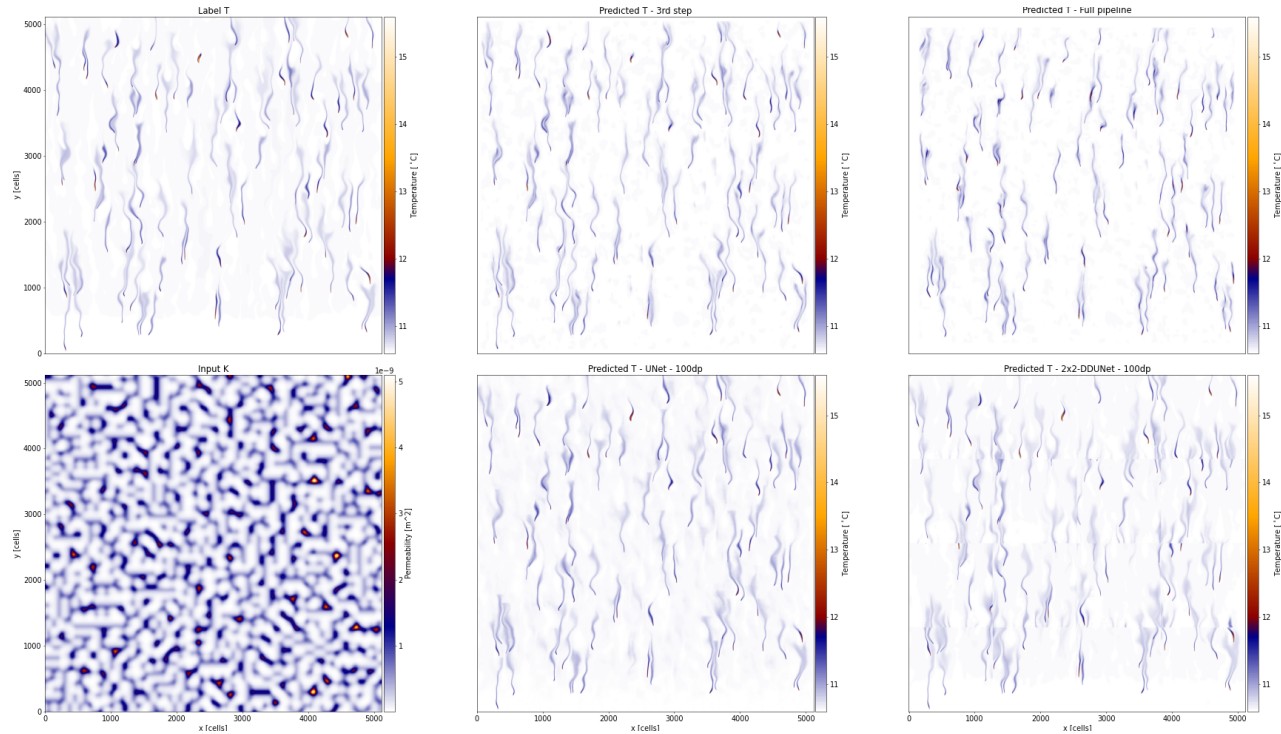

Figure 4: $pki \to T$ (baseline dataset (*3dp*), scaling). 1st column: Label $T$ and input $k$. 2nd-3rd column: Comparison of LGCNN (isolated Step 3 and full pipeline) and vanilla data-driven architectures (UNet$_{101dp}$ and $2 \times 2$-DDUNet$_{101dp}$).

**Modifying Streamline Inputs of Step 3**    We evaluate the influence of Step 3's inputs, focusing on the effect of modifying or omitting individual components of the streamlines $\vec{s} = (s, s_o)$. When both the central streamlines $s$ and offset streamlines $s_o$ are excluded, predicted plumes fail to follow flow paths beyond the CNN's receptive field. Omitting only $s_o$ results in overly narrow plumes, whereas excluding only $s$ produces overly smeared temperature fields. Including both but removing the fading of values along the streamlines leads to diffuse background temperatures and less localized plumes. Overall, the Huber validation loss increases by 32–132%, depending on the experiment. For visual, quantitative, and extended results, we refer to Appendix A.3.

## 7    DOMAIN TRANSFER TO REAL PERMEABILITY FIELDS

Finally, we investigate whether our approach transfers to the realistic dataset in Section 3, which is based on real-world permeability fields $k$ that are cut from geological maps of Munich. Since these maps cover a limited area, the number of potential datapoints is very small. We compare performance for temperature prediction of Step 3 and the full pipeline on the validation and scaling datapoints. Results are presented in Tables 4 and 5; additional metrics are in Appendix A.3.

Table 4: Metrics for predicting $\vec{v}_{\text{pred}}$ in Step 1 on validation and scaling data from our realistic dataset. In [m/y], except for MSE in [m²/y²], SSIM unitless.

| Output | Case | $L_\infty$ | MAE | MSE | SSIM |
|--------|------|------|------|------|------|
| **Step 1** | | | | | |
| $v_x$ | val | 106.8607 | 15.4095 | 380.3762 | 0.9939 |
| $v_y$ | val | 74.5570 | 10.6605 | 148.2475 | 0.9993 |
| $v_x$ | scaling | 459.1620 | 110.0078 | 13570.0000 | 0.9462 |
| $v_y$ | scaling | 240.9406 | 17.6051 | 616.5186 | 0.9965 |

Table 5: Metrics for predicting $T$ in Step 3 and in the full pipeline on validation and scaling data from our realistic dataset. In [°C], except for MSE in [°C²], PAT in [%], SSIM unitless.

| Case | $L_\infty$ | MAE | MSE | PAT | SSIM |
|------|------|------|------|------|------|
| **Step 3**, i.e., test on $\vec{v}_{\text{sim}}$ | | | | | |
| val | 0.8222 | 0.0175 | 1.01e-3 | 2.18 | 0.9672 |
| scaling | 0.8052 | 0.0189 | 8.21e-4 | 0.92 | 0.9497 |
| **Full pipeline**, i.e., test on $\vec{v}_{\text{pred}}$ | | | | | |
| val | 2.3194 | 0.0841 | 2.75e-2 | 27.79 | 0.7510 |
| scaling | 2.0511 | 0.0394 | 4.42e-3 | 10.02 | 0.8708 |

**Methodological Adaptations**   Slight adjustments to chunk size, overlap, data split, and architecture yield better performance compared to those used on the baseline dataset, see Appendix A.2.1 for details. Visual inspection of input $k$ in Figure 5 reveals fewer but larger-scale features. Therefore, training can benefit from a larger spatial context, which requires more training data. We increase the number of training samples by using three out of the four available datapoints for training and one for validation. For the streamlines computation, we switch to an explicit fourth-order Runge–Kutta scheme, which proved stable and sufficient for training and inference due to the lower frequency in $k$.

**Performance and Scaling Tests**   In Step 3, we reach an even lower maximum absolute error $L_\infty$ of 0.82 °C compared to 2.90 °C of our initial model on the baseline dataset (Section 6). Other metrics confirm a similar performance; see Table 3. In the scaling test, most errors are comparable too. For the full pipeline, errors are slightly higher on the validation and scaling data compared to an isolated Step 3, consistent with our observations on the baseline dataset (Table 3). Physical consistency, which is our primary objective, remains strong: Shape, magnitude, and connectivity of the predicted heat plumes are well preserved, even in the scaling test (Figure 5). Local deviations are primarily due to differences between $\vec{v}_{\text{sim}}$ and $\vec{v}_{\text{pred}}$, as evident from the discrepancy between Step 3 and full pipeline outputs.

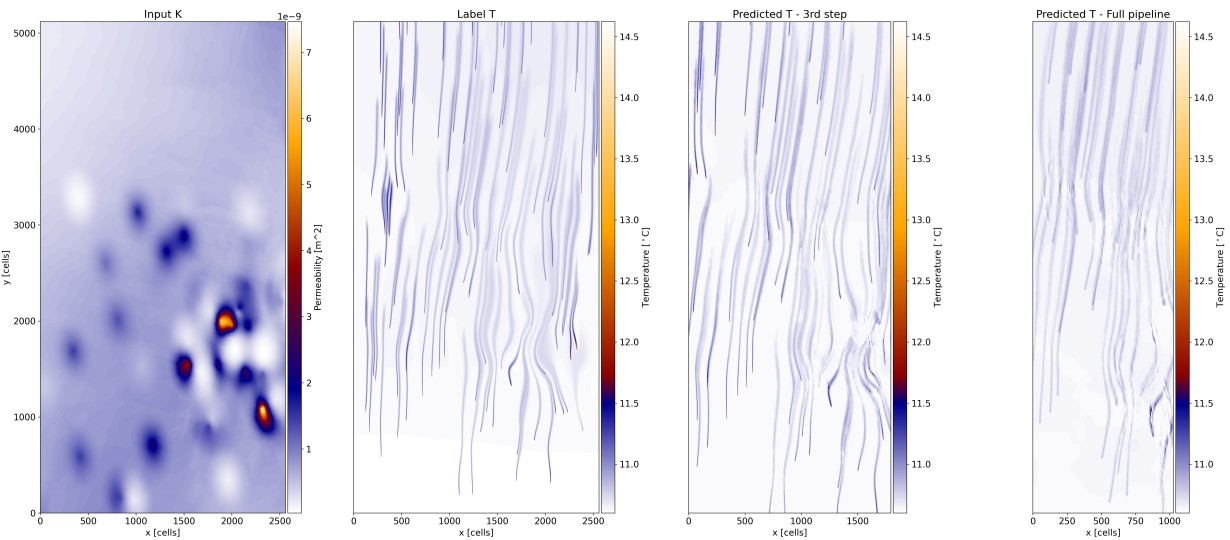

Figure 5: $pki \to T$ (realistic dataset, scaling). 1st & 2nd column: Label $T$ and input $k$. 3rd-4th column: LGCNN (isolated Step 3 and full pipeline).

## 8 Conclusions and Future Work

In real-world scientific applications, data is often very limited. At the same time, models are expected to generalize and scale to large domain sizes, pushing standard data-driven models to their limits. To address this issue, we proposed the LGCNN model, a hybrid, physics-inspired CNN approach. In the prediction of groundwater temperature fields under arbitrary heat pump configurations and real permeability data, we identify long-range interactions driven by advection as the problematic process for classical deep learning models, such as vanilla CNNs. We replace this part of the prediction with a simple numerical solver, resulting in a sequence (CNN 1 – numerical solver – CNN 2). While this setup is specific to groundwater heat transport, the general approach should be applied to a wider range of applications. Finally, we stress that traditional error metrics can obscure effects like input sensitivity; therefore, we additionally rely on visual assessments and problem-specific metrics such as PAT. In future work, we will address limitations through dataset and respective code extensions to 3D and transient states, investigate input sensitivity, and accelerate streamline computation.

ACKNOWLEDGMENTS

*Acknowledgments will be included in the final version after double-blind peer review.*

REPRODUCIBILITY

To ensure reproducibility, our code to train and evaluate models, the trained and evaluated models themselves, and the datasets used for training and testing are included in the supplementary material and will be published on Github and Dataverses after the review process.

USE OF LARGE LANGUAGE MODELS (LLMS)

The methods and research presented in the paper do not involve the use of LLMs; only for writing, editing, or formatting purposes.

DATA AND CODE ACCESSIBILITY

*Links will be included in the final version after double-blind peer review.*

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

## A  APPENDIX

### A.1  SIMULATION SETUP AND TRANSPORT REGIME

In Section 3, we describe how the two datasets are generated using the subsurface simulation software Pflotran (Lichtner et al., 2015a), which solves the coupled mass and energy conservation equations. Here, we provide additional technical details and modeling assumptions relevant for reproducibility. Furthermore, we show that for the selected parameters, the heat transport in our system is dominated by advection by a theoretical analysis.

**Hydro-geological, Operational and Simulation Parameters**  Our data was generated on 12.8km×12.8km×5m with a cell size of 5m×5m×5m for the baseline simulations. To test scalability, we also simulate a domain that is twice as large in both $x$- and $y$-dimension for the baseline dataset, but only twice as large in $y$-dimension for the more realistic dataset due to dimension restrictions of the available input data.

The *baseline dataset* uses a constant hydraulic pressure gradient $\nabla p$ of 0.003 (Pelzer & Schulte, 2024; geo, 2022); permeability field $k$ is generated using Perlin noise (Perlin, 1985) within (1.02E-11, 5.10E-9) m$^2$, and 100 randomly placed heat pumps, which are all operating with a 5 i.e., C injection temperature difference compared to the surrounding groundwater and an injection rate of 0.00024 m$^3$/s. All values are based on realistic parameter ranges in the region of Munich(geo, 2022; Zosseder et al., 2022).

For the *realistic dataset*, instead of random permeability fields $k$, we use permeability field data that is derived from borehole measurements in the Munich region(Bayerisches Landesamt für Umwelt, 2015). Furthermore, based on subsurface measurements, we set $\nabla p = 0.0025$ for these simulations. All other simulation parameters remain identical to the baseline. Other subsurface and operational parameters are taken directly from Pelzer & Schulte (2024), which also includes additional information about solver setup and boundaries. For mathematical details of the governing equations, we refer the reader to Lichtner et al. (2015b); Anderson (2007); Delleur (2016).

**Simplifications**  For simplifications, we refer to the (hydraulic) pressure field $p$ throughout the paper. In our paper, this field is the initial pressure field defined through the measured hydraulic head and is not the true pressure field at quasi-steady state with spatial details resulting from permeability $k$ variations in the domain and mass injections at the heat pump positions. The true pressure field is only available after simulation (which we are avoiding for our approach) and strongly interacts with the velocity field.

**Péclet Number**  To quantify whether heat transport in our system is dominated by advection or diffusion, we compute the dimensionless Péclet number (Rapp, 2017), which is defined as

$$\mathrm{Pe} = \frac{L \cdot v}{\alpha},$$

with the characteristic length $L$, the local velocity $v$, and the thermal diffusivity $\alpha$, defined as

$$\alpha = \frac{k}{\rho c_p},$$

with $k$ the thermal conductivity, $\rho$ the density, and $c_p$ the specific heat capacity. We take the parameters of the solid phase of our simulation inputs of

- Thermal conductivity: $k = 0.65\,\mathrm{W/(m \cdot K)}$
- Density: $\rho = 2800\,\mathrm{kg/m^3}$
- Specific heat capacity: $c_p = 2000\,\mathrm{J/(kg \cdot K)}$

This yields a thermal diffusivity of:

$$\alpha = \frac{0.65}{2800 \cdot 2000} \approx 1.16 \times 10^{-7}\,\mathrm{m^2/s}$$

The velocity values are derived by simulations, taken in the direction of flow (longitudinal) from the realistic $k$-dataset:

- Maximum: $1200\,\mathrm{m/year} \approx 3.8 \times 10^{-5}\,\mathrm{m/s}$
- Minimum: $44\,\mathrm{m/year} \approx 1.4 \times 10^{-6}\,\mathrm{m/s}$
- Mean: $330\,\mathrm{m/year} \approx 1.04 \times 10^{-5}\,\mathrm{m/s}$

With a characteristic length determined by the heat plume length of 2000–6000 m and mean x-velocity, we get:

$$Pe = \frac{L \cdot v}{\alpha} \approx L \cdot 10^2 \approx 10^5$$

The interpretation of the Peclet number is given by the following:

- $Pe \ll 1$: Diffusion dominates
- $Pe \approx 1$: Diffusion and advection similarly dominate
- $Pe \gg 1$: Advection dominates

Therefore, Pe $\approx 10^5$ indicates that in the simulations, the heat transport is advection-dominated at larger scales.

## A.2 Neural Networks, Hyperparameters and Training Details

This section provides implementation and training details for all neural network models used in this work. We begin with a description of the baseline UNet architecture, which also forms the foundation of both LGCNN and DDU-Net. Then, we outline the hyperparameter optimization process and present the final configurations per model and dataset. All models are trained using PyTorch (Paszke et al., 2017).

**UNet Architecture**   The UNet architecture used in this work is based on the original design presented by Ronneberger et al. (2015b), with several modifications to tailor it to the specific needs of our task of predicting temperature fields. In this section, we introduce the key concepts that define the architecture and explain how they contribute to the model's design. These concepts will be employed in the hyperparameter search to identify the most suitable configuration, taking into account memory and data limitations.

The UNet architecture essentially consists of a series of UNet blocks. Each block consists of the layers of "Convolution - Activation - Convolution - Norm - Activation - Convolution - Activation" with convolutional layers, a batch normalization layer, and activation functions. After each block, either an upsampling or a downsampling operation is applied via "Max Pooling 2D" or "Transposed Convolution 2D" with stride 2. The *depth* of the UNet refers to the number of UNet blocks in both the encoder and decoder. The *number of initial features* refers to the number of feature maps generated by the first downsampling block. Each downsampling block produces twice as many output feature maps as input feature maps, while each upsampling block reduces the number of feature maps by half. The *number of convolutions per block* denotes how many convolutional layers are applied within each block, while *kernel size* specifies the size of the kernels used in the convolutional operations. Additionally, we explore different activation functions (ReLU, tanh, sigmoid, and LeakyReLU) and various normalization strategies (batch normalization, group normalization, and no normalization). The UNet block exists in 2 variants, depending on the hyperparameter *repeat inner*: If *repeat inner* = False, the block looks like this "Convolution - Norm - Activation", if it is True, the block looks as described above.

The training process also involves several hyperparameters. The learning rate controls how quickly the model adjusts its parameters during training. The weight decay parameter helps prevent overfitting by penalizing large weights in the model. Furthermore, the Adam optimizer (Kingma, 2014) is employed as *optimizer*. For the realistic permeability field, we additionally introduce the optimizable hyperparameter *optimizer switch*, which, when enabled, switches the optimizer to LBFGS after 90 epochs.

During inference, each datapoint is processed as a whole, but during training, they are loaded in smaller patches cut out from the datapoint. Optimized hyperparameters include their overlap, i.e., inverse of *skip per direction*, and their size, called *box length*. The data split is untouched by this, i.e. one simulation run per train / val / test separately. This is important to not overlap test patches with training ones.

### A.2.1 Hyperparameters and Hyperparameter Optimization

We optimize the hyperparameters of our architecture, training process and data loading in several rounds with Optuna (Akiba et al., 2019) and additional manual tweaking. Optuna performs optimization using a tree-structured Parzen estimator algorithm. An overlook of all considered hyperparameters, their ranges and our final choice, as well as the hyperparameters fixed during optimization is provided per used model architecture and dataset, e.g., purely data-driven approaches, LGCNN on individual steps or the full pipeline, on datasets of random $k$ versus realistic $k$.

**Vanilla UNet**   The vanilla UNet$_{3dp}$ trained using the following hyperparameters: a batch size of 20, kernel size of 5, and a network depth of 3. The number of initial features was set to 32, with a stride and dilation of 1. We used the ReLU

activation function in combination with batch normalization. The inner block was not repeated (`repeat_inner = False`). No cutouts are applied during training (`bool_cutouts = False`), i.e., the model is trained on the whole datapoint at once.

The inputs to the model are $pki$ (pressure field, permeability field, and location of heat pumps). For training, the Mean Absolute Error (MAE) loss function was used, and optimized with the Adam optimizer. The learning rate is set to $1 \times 10^{-5}$. The model is trained for 10,000 epochs.

**Hyperparameters for purely data-driven approaches**   The values considered during hyperparameter optimization for the UNet and DDUNet, trained on 73 datapoints and performed using Optuna (Akiba et al., 2019), along with the best settings found, are listed in Table 6. Certain hyperparameters were fixed: we set the number of epochs to 750, with an early stopping criterion based on validation loss and a patience of 80 epochs. Additionally, we note that some hyperparameter combinations (e.g., 32 initial features, depth 6, and 3 convolutions per layer with a kernel size of 7) caused memory issues, leading to their exclusion from the hyperparameter search.

Table 6: Overview of used hyperparameters for the $UNet_{101dp}$ and $2 \times 2$ $DDUNet_{101dp}$ their search ranges (if applicable), and best values across training stages. Note that the number of communicated feature maps in the vanilla UNet is simply an extra convolution layer in the coarsest part of the UNet (without communication).

| Hyperparameter | Range | $pki \to v_x v_y$ | (Step 1) | $kiv_x v_y ss_o \to T$ | (Step 3) | $pki \to T$ | (Full) |
|---|---|---|---|---|---|---|---|
| | | $UNet_{101dp}$ | $2 \times 2DDUNet_{101dp}$ | $UNet_{101dp}$ | $2 \times 2DDUNet_{101dp}$ | $UNet_{101dp}$ | $2 \times 2DDUNet_{101dp}$ |
| *Dataset* | | | | | | | |
| Batch size (train) | 4, 6, 8 | 4 | 6 | 6 | 4 | 6 | 6 |
| Include pressure field | True, False | False | False | - | - | False | False |
| *Encoder-decoder properties* | | | | | | | |
| Depth | 4, 5, 6 | 6 | 5 | 6 | 6 | 5 | 5 |
| No. initial features | 8, 16, 32 | 8 | 16 | 8 | 8 | 8 | 16 |
| No. convs. per block | 1, 2, 3 | 1 | 1 | 3 | 3 | 3 | 3 |
| Kernel size | 3, 5, 7 | 7 | 7 | 3 | 5 | 7 | 7 |
| *Communication Network* | | | | | | | |
| No. comm. feature maps | 64, 128, 256 | 64 | 128 | 256 | 64 | 256 | 256 |
| *Training* | | | | | | | |
| Learning rate | [1e-5, 1e-3] | 0.00024 | 0.00100 | 0.00017 | 0.00030 | 0.00024 | 0.00024 |
| Weight decay | 0.0, 0.001 | 0.0 | 0.0 | 0.0 | 0.0 | 0.0 | 0.0 |
| Train loss | MSE, L1 | MSE | MSE | L1 | MSE | MSE | MSE |

After the hyperparameter search, the values corresponding to the best-performing model (based on Huber loss for the validation dataset) were selected. With these values fixed, five models were trained using different randomly sampled initializations to evaluate sensitivity to random initialization, for these values see Tables 10 and 11.

**LGCNN Hyperparameters - on Random Permeability**   The values considered during hyperparameter optimization with Optuna (Akiba et al., 2019) and the best settings found for both steps of LGCNN are listed in Table 7. Although the optimization was originally run for 100 epochs, the optimum was consistently found within the first 25 epochs. Therefore, to reduce computation cost, we therefore conservatively lowered the maximum number of epochs to 50. This adjustment does not affect any of the reported metrics in the paper.

Fixed parameters for this hyperparameter search are the learning rate (fixed at $10^{-4}$), ReLU as activation function, the batch size of 20, and the use of a batch normalization layer within the inner blocks of the UNet architecture. The validation loss used for selecting the optimal model is the MAE.

**LGCNN Hyperparameters - on Real Permeability**   The values considered during hyperparameter optimization on the dataset with a more realistic permeability field were selected using Optuna (Akiba et al., 2019), and are summarized in Table 8, along with the best configurations found for both steps of the LGCNN. The optimization was run for up to 100 epochs. For more background on the network architecture and the various hyperparameters, cf. A.2. Several hyperparameters were fixed during this process. These include a constant learning rate schedule, an Adam optimizer with a weight decay of $10^{-4}$, and, when enabled, a switch to LBFGS after 90 epochs. Fixed architectural parameters

Table 7: LGCNN-Random $k$: Hyperparameter optimization: Parameter ranges and best configurations.

| Parameter | Range | 1st step | 3rd step |
|---|---|---|---|
| *Parameters of the Dataset* | | | |
| inputs | $v : (p, i, k)$ $T : (i, v_x, v_y, s, s_o, k)$ | $pik$ | $iv_x v_y ss_o k$ |
| skip per direction | $v : 4, 8, 16, 32, 64$ $T : 8, 16, 32, 64$ | 16 | 8 |
| box length | 64, 128, 256, 512 | 256 | 256 |
| *Parameters of Training* | | | |
| loss function (training) | MAE, MSE | MSE | MAE |
| optimizer | Adam, SGD | Adam | Adam |
| *Parameters of the Network* | | | |
| No. initial features | 8, 16, 32, 64, 128 | 32 | 32 |
| kernel size | 3, 4, 5 | 5 | 4 |
| depth | $v : 1, 2, 3, 4$ $T : 1, 2, 3$ | 4 | 4 |

include a convolutional stride and dilation of 1. During training, the inputs were cut out from the full datapoints. For model comparison, the validation loss was consistently computed using the Huber loss.

## A.3 ADDITIONAL EXPERIMENTAL RESULTS

This section provides additional experimental results. While the main results section focused only on the test and scaling datasets, we also include here the metric values on the training and validation datasets. For completeness and easier comparison, the test and scaling metrics are re-listed as well.

**Purely data-driven UNet, DDU-Net: Metrics of training and ablation study**  Table 9 presents all the metrics for predicting the temperature field directly from the inputs $pki$ using a data-driven approach, evaluated on the training, validation, and test datasets. The results are provided for several models: (1) UNet trained on only 1 datapoint and tested and validated on 2 additional datapoints, (2) a UNet trained on 73 datapoints (73-18-10 train-validation-test split), and (3) a DDUNet trained on the same 73 datapoints dataset, operating on $2 \times 2$ subdomains. Furthermore, to assess the model's sensitivity to random initialization, the training of the same architecture was repeated five times for the most relevant models. Based on these repetitions, the mean and standard deviation of the performance metrics were computed using the following equations:

$$\bar{x} = \frac{1}{n} \sum_{i=1}^{n} x_i \quad \text{and} \quad \sigma = \sqrt{\frac{1}{n-1} \sum_{i=1}^{n} (x_i - \bar{x})^2}$$

where $x_i$ denotes the metric value from the $i$-th training run, and $n = 5$ is the number of runs. These results are summarized in Table 10 and Table 11. The choice of $n = 5$ was made empirically to balance computational effort and statistical reliability. The standard deviations in  Tables 10 and 11 were used as validation: they are neither excessively large (indicating instability) nor unrealistically small (indicating insufficient sampling).

In addition to testing the UNet and DDUNet trained on 73 datapoints on the $101_{dp}$ test dataset, we also evaluate these models on the same datapoint used to test the UNet$_{3DP}$.

**LGCNN: Metrics of training and ablation study**  The results of the LGCNN and DDU-Net, evaluated on both synthetic and realistic permeability fields, for the training, validation, testing, and scaling datasets are shown in Table 12

Table 8: LGCNN-Real $k$: Hyperparameter optimization: Parameter ranges and best configurations.

| Parameter | Range | model $\vec{v}$ | model $T$ |
|---|---|---|---|
| *Parameters of the Dataset* | | | |
| inputs | $v : ki, pik, gik, gk, pk$ | $pik$ | |
| | $T : iv_x v_y ss_o k$ | | $iv_x v_y ss_o k$ |
| batch size | 2, 4, 8, 16 | 8 | 8 |
| skip per direction | 256, 128, 64, 32, 16, 8 | 8 | 8 |
| box length | 1280, 640 | 1280 | 1280 |
| *Parameters of Training* | | | |
| loss function (training) | MSE, MAE | MSE | MSE |
| optimizer switch | True, False | False | False |
| learning rate | 1e-3, 5e-4, 1e-4, 5e-5 | 1e-4 | 1e-4 |
| *Parameters of the Network* | | | |
| No. initial features | 8, 16, 32 | 16 | 16 |
| kernel size | 3, 5 | 5 | 5 |
| depth | 4, 5, 6 | 6 | 6 |
| repeat inner | True, False | False | False |
| activation function | relu, tanh, sigmoid, leakyrelu | relu | relu |
| layer norm | batch-, group-, None | batch- | batch- |

Table 9: Performance metrics for predicting $T$ with different models and datasets. Errors in [°C], MSE in [°C], PATs in [%] and SSIM unitless. The LGCNN-test dataset corresponds to the 1 datapoint used for testing the LGCNN approach.

| Model | Data | Case | Huber | $L_\infty$ | MAE | MSE | PAT | SSIM |
|---|---|---|---|---|---|---|---|---|
| UNet$_{3DP}$ | randomK3 | train | 0.0020 | 2.4954 | 0.0404 | 0.0040 | 6.43 | 0.8281 |
| | | val | 0.0269 | 5.2901 | 0.1365 | 0.0574 | 38.82 | 0.5717 |
| | | test | 0.0235 | 4.8642 | 0.1314 | 0.0492 | 39.05 | 0.5794 |
| UNet | rK101 | train | 0.0010 | 4.2443 | 0.0172 | 0.0021 | 2.34 | 0.9960 |
| | | val | 0.0051 | 4.1972 | 0.0441 | 0.0106 | 12.68 | 0.9859 |
| | | test | 0.0050 | 4.2140 | 0.0426 | 0.0104 | 11.90 | 0.9869 |
| | * | LGCNN-test | 0.0048 | 4.3985 | 0.0473 | 0.0100 | 13.63 | 0.9827 |
| | * | scaling | 0.0016 | 4.3426 | 0.0202 | 0.0033 | 4.33 | 0.9955 |
| 2×2-DDUNet | rK101 | train | 0.0018 | 3.8100 | 0.0236 | 0.0038 | 4.47 | 0.9940 |
| | | val | 0.0079 | 4.0084 | 0.0550 | 0.0165 | 17.03 | 0.9825 |
| | | test | 0.0076 | 3.7006 | 0.0549 | 0.0159 | 17.14 | 0.9835 |
| | * | LGCNN-test | 0.0063 | 3.4257 | 0.0548 | 0.0128 | 17.42 | 0.9804 |
| | * | scaling | 0.0025 | 4.1806 | 0.0235 | 0.0052 | 6.11 | 0.9940 |

Table 10: Statistics for predicting $T$ with different models and datasets. Errors in [°C], MSE in [°C], PATs in [%] and SSIM unitless. The LGCNN-test dataset corresponds to the 1 datapoint used for testing the LGCNN approach. Mean ± standard deviation reported.

| Model | Data | Case | Huber | $L_\infty$ | MAE | MSE | PAT | SSIM |
|---|---|---|---|---|---|---|---|---|
| UNet$_{3dp}$ | randomK3 | train | $0.0025 \pm 0.0024$ | $4.69 \pm 0.17$ | $0.0155 \pm 0.0071$ | $0.0063 \pm 0.0062$ | $1.92 \pm 1.16$ | $0.981 \pm 0.012$ |
| | | val | $0.0219 \pm 0.0008$ | $4.89 \pm 0.07$ | $0.1114 \pm 0.0023$ | $0.0474 \pm 0.0019$ | $34.76 \pm 0.99$ | $0.696 \pm 0.006$ |
| | | test | $0.0176 \pm 0.0006$ | $4.85 \pm 0.09$ | $0.1043 \pm 0.0018$ | $0.0368 \pm 0.0015$ | $34.76 \pm 0.76$ | $0.703 \pm 0.006$ |
| UNet$_{101dp}$ | rK101 | train | $0.0011 \pm 0.0002$ | $4.37 \pm 0.20$ | $0.0182 \pm 0.0020$ | $0.0023 \pm 0.0004$ | $2.45 \pm 0.27$ | $0.995 \pm 0.002$ |
| | | val | $0.0055 \pm 0.0003$ | $4.36 \pm 0.16$ | $0.0454 \pm 0.0019$ | $0.0114 \pm 0.0006$ | $12.96 \pm 0.55$ | $0.984 \pm 0.002$ |
| | | test | $0.0052 \pm 0.0003$ | $4.35 \pm 0.23$ | $0.0441 \pm 0.0019$ | $0.0110 \pm 0.0006$ | $12.49 \pm 0.48$ | $0.985 \pm 0.002$ |
| | * | LGCNN-test | $0.0049 \pm 0.0002$ | $4.30 \pm 0.16$ | $0.0470 \pm 0.0010$ | $0.0102 \pm 0.0004$ | $13.51 \pm 0.59$ | $0.983 \pm 0.002$ |
| | * | scaling | $0.0017 \pm 0.0001$ | $4.48 \pm 0.15$ | $0.0208 \pm 0.0014$ | $0.0035 \pm 0.0002$ | $4.38 \pm 0.17$ | $0.995 \pm 0.001$ |
| 2×2-DDUNet$_{101dp}$ | rK101 | train | $0.0014 \pm 0.0003$ | $4.11 \pm 0.25$ | $0.0203 \pm 0.0026$ | $0.00300 \pm 0.0007$ | $3.24 \pm 0.68$ | $0.995 \pm 0.001$ |
| | | val | $0.0079 \pm 0.0002$ | $4.20 \pm 0.25$ | $0.0564 \pm 0.0008$ | $0.01648 \pm 0.0005$ | $17.32 \pm 0.22$ | $0.981 \pm 0.002$ |
| | | test | $0.0075 \pm 0.0001$ | $4.05 \pm 0.22$ | $0.0552 \pm 0.0008$ | $0.01580 \pm 0.0002$ | $16.94 \pm 0.36$ | $0.982 \pm 0.001$ |
| | * | LGCNN-test | $0.0057 \pm 0.0003$ | $4.00 \pm 0.20$ | $0.0526 \pm 0.0015$ | $0.01171 \pm 0.0006$ | $16.44 \pm 0.63$ | $0.981 \pm 0.002$ |
| | * | scaling | $0.0025 \pm 0.0001$ | $4.04 \pm 0.20$ | $0.0251 \pm 0.0007$ | $0.00514 \pm 0.0002$ | $6.39 \pm 0.17$ | $0.994 \pm 0.001$ |
| LGCNN | randomK3 Step 3 | train | $0.0001 \pm 0.0001$ | $2.64 \pm 0.18$ | $0.0064 \pm 0.0008$ | $0.0003 \pm 0.0001$ | $0.29 \pm 0.08$ | $0.996 \pm 0.001$ |
| | | val | $0.0032 \pm 0.0001$ | $2.49 \pm 0.29$ | $0.0413 \pm 0.0008$ | $0.0065 \pm 0.0003$ | $10.88 \pm 0.35$ | $0.912 \pm 0.003$ |
| | | test | $0.0025 \pm 0.0000$ | $2.74 \pm 0.33$ | $0.0382 \pm 0.0006$ | $0.0049 \pm 0.0001$ | $9.44 \pm 0.35$ | $0.918 \pm 0.003$ |
| | | scaling | $0.0008 \pm 0.0000$ | $3.03 \pm 0.31$ | $0.0179 \pm 0.0006$ | $0.0016 \pm 0.0001$ | $2.54 \pm 0.14$ | $0.946 \pm 0.005$ |

Table 11: Statistics for predicting $\vec{v}$ with the randomK dataset. Errors in [m/y], MSE in [m²/y²], SSIM unitless. Mean ± standard deviation reported.

| Model | Data | Output | Case | Huber | $L_\infty$ | MAE | MSE | SSIM |
|---|---|---|---|---|---|---|---|---|
| LGCNN | randomK3 | $v_x$ | train | $14.11 \pm 11.52$ | $96.63 \pm 23.29$ | $14.60 \pm 11.53$ | $380.19 \pm 529.31$ | $0.997 \pm 0.002$ |
| | | $v_y$ | train | $14.30 \pm 7.10$ | $135.26 \pm 20.36$ | $14.80 \pm 7.11$ | $361.61 \pm 296.21$ | $0.991 \pm 0.006$ |
| | | $v_x$ | val | $28.23 \pm 7.76$ | $445.13 \pm 35.50$ | $28.72 \pm 7.77$ | $1601.10 \pm 587.80$ | $0.989 \pm 0.003$ |
| | | $v_y$ | val | $27.33 \pm 2.25$ | $343.35 \pm 26.80$ | $27.82 \pm 2.25$ | $1587.57 \pm 252.10$ | $0.982 \pm 0.002$ |
| | | $v_x$ | test | $26.94 \pm 5.99$ | $216.74 \pm 18.92$ | $27.44 \pm 5.99$ | $1368.52 \pm 431.48$ | $0.990 \pm 0.002$ |
| | | $v_y$ | test | $30.38 \pm 6.18$ | $249.11 \pm 11.37$ | $30.88 \pm 6.18$ | $1848.47 \pm 637.10$ | $0.982 \pm 0.005$ |
| | | $v_x$ | scaling | $27.89 \pm 6.53$ | $286.08 \pm 22.96$ | $28.38 \pm 6.53$ | $1486.23 \pm 451.52$ | $0.990 \pm 0.002$ |
| | | $v_y$ | scaling | $27.60 \pm 2.43$ | $328.80 \pm 24.95$ | $28.09 \pm 2.43$ | $1578.87 \pm 258.66$ | $0.979 \pm 0.003$ |
| LGCNN | realK | $v_x$ | train | $29.66 \pm 2.29$ | $592.17 \pm 37.66$ | $30.16 \pm 2.29$ | $2326.33 \pm 327.11$ | $0.988 \pm 0.005$ |
| | | $v_y$ | train | $28.93 \pm 5.47$ | $397.09 \pm 72.78$ | $29.43 \pm 5.47$ | $1778.28 \pm 551.97$ | $0.997 \pm 0.009$ |
| | | $v_x$ | val | $22.17 \pm 2.24$ | $170.47 \pm 9.75$ | $22.66 \pm 2.24$ | $909.39 \pm 139.94$ | $0.988 \pm 0.002$ |
| | | $v_y$ | val | $20.20 \pm 4.07$ | $113.33 \pm 12.37$ | $20.70 \pm 4.08$ | $645.11 \pm 221.50$ | $0.997 \pm 0.001$ |
| | | $v_x$ | scaling | $79.33 \pm 21.80$ | $532.35 \pm 140.50$ | $79.83 \pm 21.80$ | $9404.62 \pm 4048.09$ | $0.953 \pm 0.014$ |
| | | $v_y$ | scaling | $53.39 \pm 12.64$ | $602.41 \pm 128.79$ | $53.89 \pm 12.64$ | $4621.25 \pm 1755.50$ | $0.980 \pm 0.006$ |

(for Step 1 - predict $\vec{v}_{\text{pred}}$) and Table 13 (for Step 3 - predict $T$). The column "data" refers to the dataset that a model was trained on and applied to, i.e., "randomK" stands for the baseline dataset *3dp*, "randomK101" for *101dp*. An asterisk indicates that the model was evaluated on a dataset different from the one it was trained on. Mostly relevant for the vanilla approaches that are trained on *101dp* and applied to scaling and test of *3dp*.

Table 12: Performance metrics for predicting $\vec{v}$ with different models and datasets. Errors in [m/y], MSE in [m$^2$/y$^2$], SSIM unitless.

| Model | Data | Output | Case | Huber | $L_\infty$ | MAE | MSE | SSIM |
|---|---|---|---|---|---|---|---|---|
| 1st Step | | | | | | | | |
| | | $v_x$ | train | 9.4732 | 132.6672 | 9.9620 | 171.0032 | 0.9972 |
| | | $v_y$ | train | 11.4005 | 223.4601 | 11.8902 | 275.6190 | 0.9937 |
| LGCNN | randomK | $v_x$ | val | 22.2241 | 343.9907 | 22.7179 | 1102.3721 | 0.9905 |
| | | $v_y$ | val | 26.6078 | 274.8036 | 27.1026 | 1524.5099 | 0.9841 |
| | | $v_x$ | test | 21.8237 | 190.8046 | 22.3178 | 972.5668 | 0.9911 |
| | | $v_y$ | test | 32.2488 | 256.2519 | 32.7444 | 2031.3357 | 0.9812 |
| * | | $v_x$ | scaling | 24.4314 | 294.0457 | 24.9261 | 1204.1154 | 0.9911 |
| * | | $v_y$ | scaling | 25.7847 | 367.6891 | 26.2795 | 1463.8218 | 0.9820 |
| Experiment: trained on full image | | | | | | | | |
| | | $v_x$ | train | 3.1983 | 39.9380 | 3.6714 | 21.5058 | 0.9993 |
| | | $v_y$ | train | 3.0496 | 43.9018 | 3.5177 | 21.4905 | 0.9988 |
| LGCNN | randomK | $v_x$ | val | 25.2418 | 417.1447 | 25.7367 | 1421.6547 | 0.9896 |
| | | $v_y$ | val | 29.6294 | 276.8029 | 30.1247 | 1827.3134 | 0.9824 |
| | | $v_x$ | test | 23.3646 | 202.6397 | 23.8584 | 1144.4479 | 0.9913 |
| | | $v_y$ | test | 35.4289 | 332.9557 | 35.9249 | 2490.4104 | 0.9789 |
| Replace with UNet-101dp, i.e., $1 \times 1$ subdomain-DDUNet, trained on 101 datapoints | | | | | | | | |
| | | $v_x$ | train | 3.5468 | 678.7761 | 4.0134 | 86.6731 | 0.9767 |
| | | $v_y$ | train | 4.1030 | 690.1506 | 4.5816 | 100.2767 | 0.9934 |
| | randomK101 | $v_x$ | val | 3.8368 | 647.1833 | 4.3058 | 90.3292 | 0.9739 |
| | | $v_y$ | val | 4.3649 | 685.6656 | 4.8433 | 102.8601 | 0.9931 |
| DDUNet | | $v_x$ | test | 3.6643 | 584.8119 | 4.1325 | 83.4062 | 0.9731 |
| | | $v_y$ | test | 4.1355 | 547.8865 | 4.6131 | 87.7819 | 0.9924 |
| * | | $v_x$ | LGCNN-test | 12.2161 | 1047.8464 | 12.7036 | 896.2758 | 0.9016 |
| * | | $v_y$ | LGCNN-test | 13.9033 | 2594.9504 | 14.3878 | 2317.6863 | 0.9903 |
| * | | $v_x$ | scaling | 4.7412 | 659.8439 | 5.2152 | 79.2508 | 0.9699 |
| * | | $v_y$ | scaling | 5.2686 | 456.1904 | 5.7492 | 83.9735 | 0.9948 |
| Replace with 2x2-DDUNet-101dp, i.e.,$2 \times 2$ subdomains | | | | | | | | |
| | | $v_x$ | train | 5.3928 | 422.4526 | 5.8815 | 57.6045 | 0.9580 |
| | | $v_y$ | train | 4.0476 | 696.2919 | 4.5116 | 58.8457 | 0.9972 |
| | randomK101 | $v_x$ | val | 5.7663 | 471.6309 | 6.2545 | 76.4372 | 0.9552 |
| | | $v_y$ | val | 4.5085 | 644.1844 | 4.9751 | 71.5679 | 0.9973 |
| DDUNet | | $v_x$ | test | 5.5847 | 319.9844 | 6.0735 | 59.2501 | 0.9526 |
| | | $v_y$ | test | 4.2292 | 535.0656 | 4.6948 | 54.5020 | 0.9972 |
| * | | $v_x$ | LGCNN-test | 13.1829 | 2453.5881 | 13.6735 | 1367.9061 | 0.9325 |
| * | | $v_y$ | LGCNN-test | 14.6481 | 2029.0269 | 15.1332 | 2221.4226 | 0.9919 |
| * | | $v_x$ | scaling | 7.6311 | 421.4186 | 8.1184 | 125.9511 | 0.9314 |
| * | | $v_y$ | scaling | 6.9406 | 400.2490 | 7.4200 | 135.7929 | 0.9955 |
| **Domain Transfer**, i.e., to more realistic data | | | | | | | | |
| | | $v_x$ | train | 13.6890 | 122.7981 | 14.1819 | 340.6713 | 0.9973 |
| | | $v_y$ | train | 9.0680 | 71.2576 | 9.5619 | 126.7222 | 0.9991 |
| LGCNN | realK | $v_x$ | val | 14.9187 | 106.8607 | 15.4095 | 380.3762 | 0.9939 |
| | | $v_y$ | val | 10.1675 | 74.5570 | 10.6605 | 148.2475 | 0.9993 |
| * | | $v_x$ | scaling | 109.5079 | 459.1620 | 110.0078 | 13570.0000 | 0.9462 |
| * | | $v_y$ | scaling | 17.1118 | 240.9406 | 17.6051 | 616.5186 | 0.9965 |

**LGCNN+random $k$: Performance of Step 1** The model generally obtains good results in Figure 3, even for cells that are far away from injection points.

Table 13: Performance metrics for predicting $T$ with different models and datasets. Errors in [$°C$], MSE in [$°C^2$], PATs in [%], SSIM unitless.

| Model | Data | Case | Huber | $L_\infty$ | MAE | MSE | PAT | SSIM |
|---|---|---|---|---|---|---|---|---|
| **3rd Step: trained and applied to $v_{sim}$** | | | | | | | | |
| LGCNN | randomK $v_{sim}$ | train | 1.87e-5 | 2.2536 | 0.0014 | 3.95e-5 | 0.03 | 0.9997 |
| | | val | 0.0027 | 2.8857 | 0.0369 | 0.0054 | 8.61 | 0.9283 |
| | | test | 0.0021 | 2.8990 | 0.0347 | 0.0041 | 7.54 | 0.9304 |
| | * | scaling | 0.0007 | 3.0250 | 0.0168 | 0.0014 | 2.05 | 0.9510 |
| **Full Pipeline, i.e., Step 3 trained on $v_{sim}$, but applied to $v_{pred}$** | | | | | | | | |
| LGCNN | randomK $v_{pred}$ | train | 0.0121 | 5.1006 | 0.0642 | 0.0272 | 18.17 | 0.8700 |
| | | val | 0.0188 | 4.2264 | 0.0967 | 0.0411 | 28.96 | 0.7625 |
| | | test | 0.0147 | 4.2120 | 0.0905 | 0.0307 | 28.92 | 0.7637 |
| | * | scaling | 0.0065 | 4.9366 | 0.0413 | 0.0141 | 10.87 | 0.8654 |
| **Experiments on 3rd Step** | | | | | | | | |
| Trained in sequence, i.e., trained on $v_{pred}$ | | | | | | | | |
| LGCNN | randomK $v_{pred}$ | train | 0.0032 | 4.0750 | 0.0198 | 0.0068 | 3.73 | 0.9794 |
| | | val | 0.0195 | 3.8178 | 0.1003 | 0.0423 | 30.46 | 0.7442 |
| | | test | 0.0139 | 4.0804 | 0.0901 | 0.0289 | 29.44 | 0.7553 |
| Trained with zero-padding | | | | | | | | |
| LGCNN | randomK $v_{sim}$ | train | 0.0002 | 1.8465 | 0.0140 | 0.0004 | 0.33 | 0.9518 |
| | | val | 0.0032 | 2.9051 | 0.0465 | 0.0064 | 11.31 | 0.8686 |
| | | test | 0.0024 | 2.3391 | 0.0418 | 0.0048 | 9.19 | 0.8838 |
| Trained on full image | | | | | | | | |
| LGCNN | randomK $v_{sim}$ | train | 0.0002 | 3.5648 | 0.0022 | 0.0004 | 0.09 | 0.9992 |
| | | val | 0.0027 | 3.5088 | 0.0375 | 0.0055 | 8.81 | 0.9217 |
| | | test | 0.0021 | 3.6420 | 0.0351 | 0.0043 | 7.41 | 0.9245 |
| Replace with UNet-101dp: Step 3 trained and applied on $v_{sim}$ | | | | | | | | |
| DDUNet | rK101 | train | 0.0019 | 3.8491 | 0.0283 | 0.0038 | 5.35 | 0.9911 |
| | | val | 0.0027 | 3.9381 | 0.0350 | 0.0055 | 8.89 | 0.9895 |
| | | test | 0.0025 | 3.7999 | 0.0343 | 0.0052 | 8.50 | 0.9903 |
| | * | LGCNN-test | 0.0032 | 4.4658 | 0.0416 | 0.0067 | 11.69 | 0.9835 |
| | * | scaling | 0.0010 | 4.3489 | 0.0176 | 0.0022 | 3.07 | 0.9954 |
| Replace with 2x2-DDUNet-101dp: Step 3 trained and applied on $v_{sim}$ | | | | | | | | |
| DDUNet | rK101 | train | 0.0015 | 4.0624 | 0.0200 | 0.0032 | 3.44 | 0.9952 |
| | | val | 0.0040 | 4.1863 | 0.0410 | 0.0083 | 12.54 | 0.9904 |
| | | test | 0.0039 | 4.0807 | 0.0411 | 0.0081 | 12.67 | 0.9910 |
| | * | LGCNN-test | 0.0029 | 4.2772 | 0.0376 | 0.0058 | 10.96 | 0.9915 |
| | * | scaling | 0.0012 | 3.9946 | 0.0192 | 0.0025 | 4.73 | 0.9968 |
| **Domain Transfer**, i.e. to more realistic data | | | | | | | | |
| 3rd Step | | | | | | | | |
| LGCNN | realK $v_{sim}$ | train | 0.0002 | 0.7704 | 0.0139 | 0.0005 | 0.43 | 2.9107 |
| | | val | 0.0005 | 0.8222 | 0.0175 | 0.0010 | 2.18 | 0.9672 |
| | * | scaling | 0.0004 | 0.8052 | 0.0189 | 0.0008 | 0.92 | 0.9497 |
| Full Pipeline | | | | | | | | |
| LGCNN | realK $v_{pred}$ | train | 0.0049 | 2.5437 | 0.0534 | 0.0100 | 17.58 | 2.4923 |
| | | val | 0.0137 | 2.3194 | 0.0841 | 0.0275 | 27.79 | 0.7510 |
| | * | scaling | 0.0022 | 2.0511 | 0.0394 | 0.0044 | 10.02 | 0.8708 |

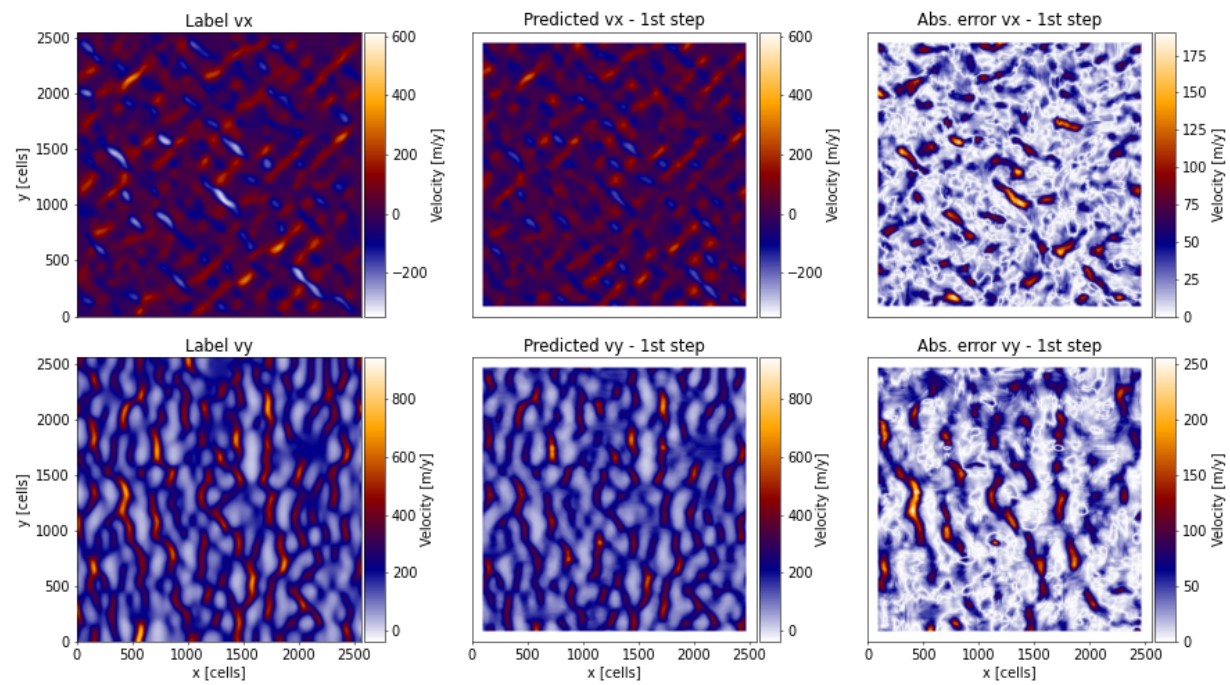

Figure 6: Prediction of $\vec{v}$ with LGCNN. Left: Test datapoint with labels $v_x$ and $v_y$. Middle: Step 1 prediction for $v_x$, $v_y$. Right: Error of the $v_x$, $v_y$ prediction.

**LGCNN+random $k$: Experiment on Inputs to Step 3**    Additional tests show that excluding other inputs, either $i$ alone or both $i$ and $k$, raises prediction error by 58–121%. We also evaluated alternative time-stepping schemes for solving the IVP. Replacing the 5th-order implicit Runge–Kutta method with explicit 2nd- or 4th-order schemes accelerates computation, but increases prediction error by 16–23%—a moderate degradation compared to the complete removal of streamline inputs. Nonetheless, we retain the implicit scheme for its superior accuracy and stability. Quantitative and qualitative results for the predictions are shown in Figure 7 and Table 14.

Table 14: Experiment on 3rd step: Test metrics for predicting $T$ with different input combinations. Errors in [°C], MSE in [°C$^2$], PAT in [%], SSIM unitless.

| Inputs | Huber | $L_\infty$ | MAE | MSE | PAT | SSIM |
|---|---|---|---|---|---|---|
| $ikv_xv_y$ | 0.0070 | 2.2990 | 0.0712 | 0.0139 | 25.55 | 0.7662 |
| $ikv_xv_ys$ | 0.0041 | 1.8674 | 0.0545 | 0.0083 | 20.56 | 0.8368 |
| $ikv_xv_ys_o$ | 0.0057 | 2.6623 | 0.0598 | 0.0114 | 20.64 | 0.8423 |
| $ikv_xv_yss_o$ (not faded) | 0.0072 | 2.3039 | 0.0744 | 0.0144 | 21.40 | 0.7837 |
| $ikv_xv_yss_o^{\mathrm{a}}$ | 0.0031 | 1.8364 | 0.0442 | 0.0062 | 12.97 | 0.8828 |
| $v_xv_yss_o$ | 0.0066 | 2.1301 | 0.0647 | 0.0132 | 20.64 | 0.8681 |
| $kv_xv_yss_o$ | 0.0049 | 2.0925 | 0.0587 | 0.0097 | 20.32 | 0.8732 |
| explicit RK, order 4 | 0.0038 | 3.2636 | 0.0486 | 0.0076 | 15.80 | 0.8871 |
| explicit RK, order 2 | 0.0036 | 2.2034 | 0.0463 | 0.0072 | 13.75 | 0.8830 |

[a] new run to be comparable to the others in this experiment: trained with Huber validation loss, hence the results differ slightly wrt. to Table 3.

**LGCNN+realistic $k$: Performance of 3rd step and full pipeline**    The qualitative performance is observable in Figure 8, where we see coherent streamlines and plume structures for both the isolated 3rd step and the full pipeline.

**Training and inference times**    Table 15 summarizes the training and inference times, number of epochs, and dataset splits (train:val:test) for each of the three steps in our pipeline, both for the LGCNN trained on partitioned and full

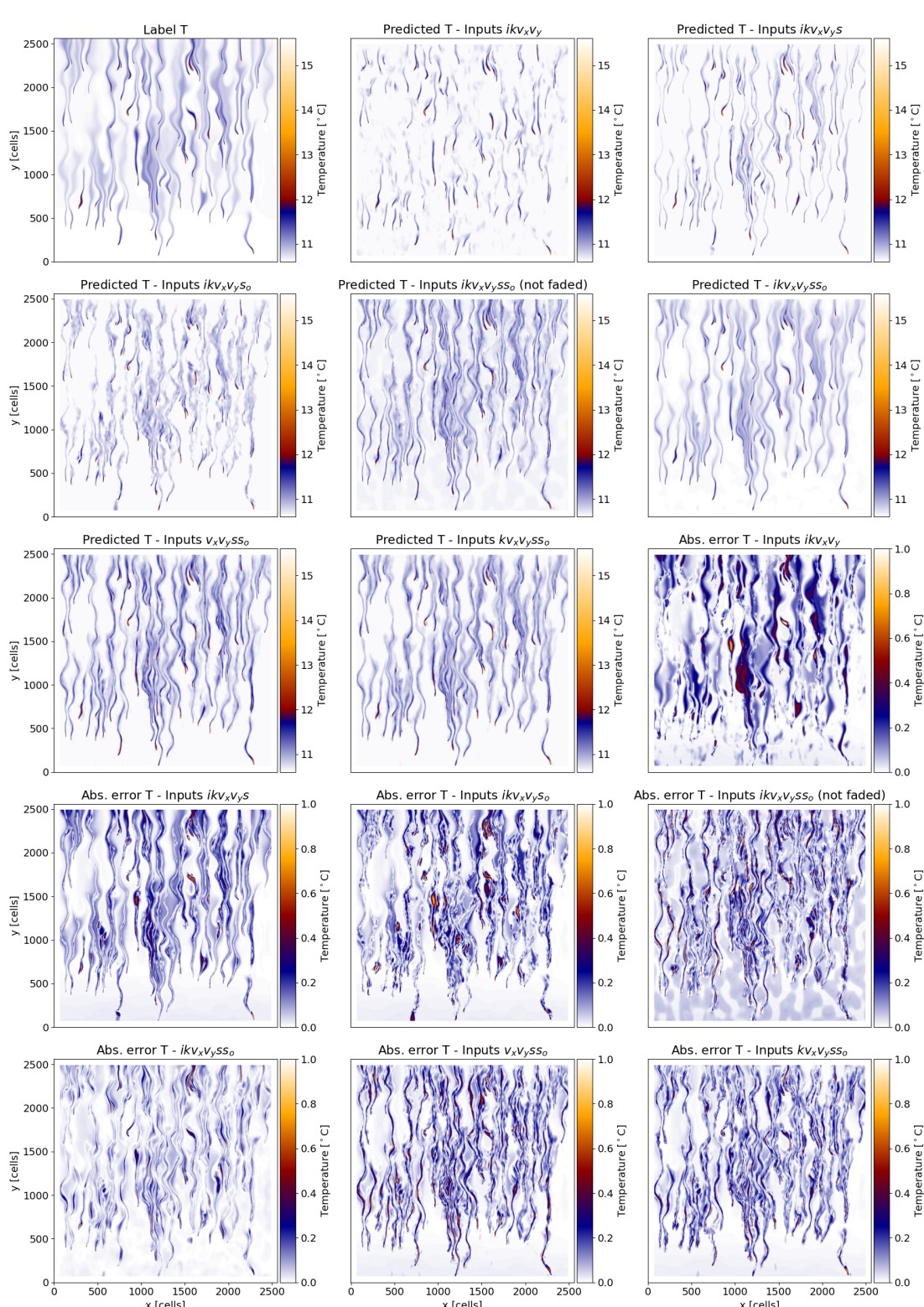

Figure 7: 1st Column: Label, input $k$, 2nd-5th: Without $(s, s_o)$, $(s_o)$, $(s)$, not-faded streamlines $(s, s_o)$, 6th: include all inputs, 7th-8th: Without $(i, k)$, $(i)$. Absolute errors capped at $1°C$ for better visualizations. Maximum errors are listed in Table 14.

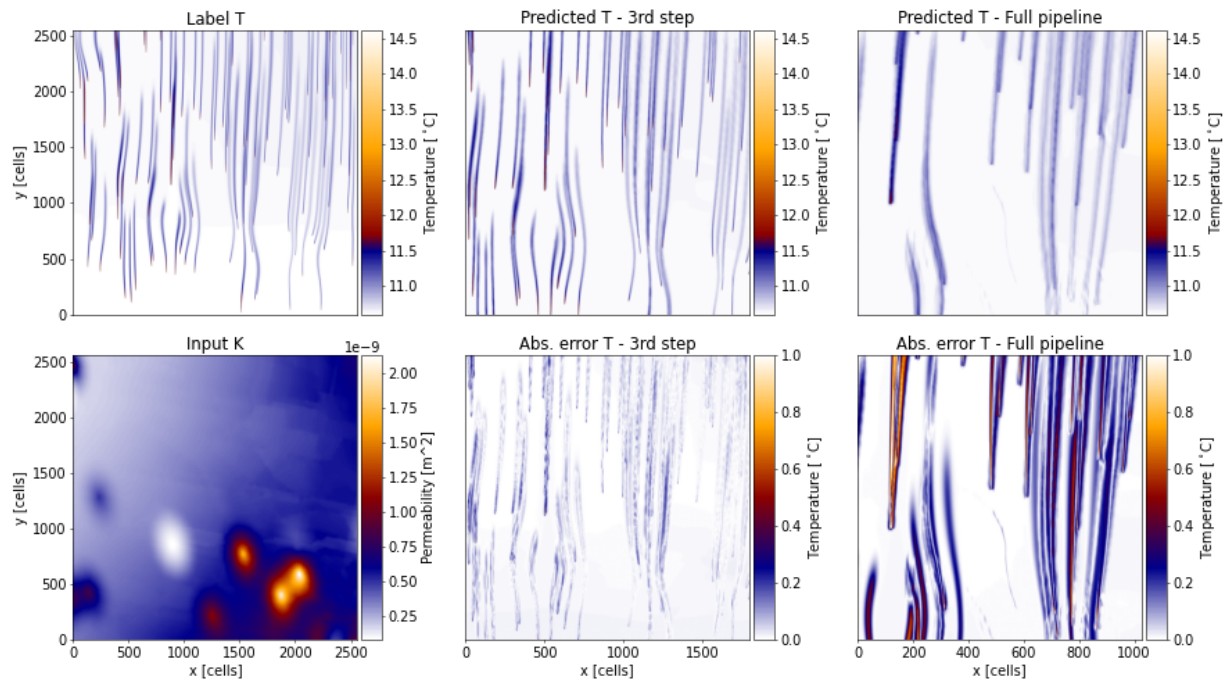

Figure 8: 1st Column: Label, input $k$. 2nd Column: 3rd step prediction of $T$ and error with respect to the label. 3rd Column: Predicted $T$ and error of the full pipeline.

datasets. Although both approaches exhibit similar inference times, they show significant differences in training time, with the partitioned approach yielding better results.

Table 15: Training measurements on the randomK dataset.

| | Data Points (train:val:test) | Epochs[a] | Training Time[a] | Inference Time |
|---|---|---|---|---|
| 1st Step (partitioned) | 20,736:1:1 | 19 | 9.5 min | 0.02 s |
| 1st Step (full) | 1:1:1 | 9,688 | 92.6 min | 0.02 s |
| 2nd Step | 1:1:1 | - | - | 9.82 s |
| 3rd Step (partitioned) | 82,944:1:1 | 14 | 31.5 min | 0.03 s |
| 3rd Step (full) | 1:1:1 | 9,671 | 92.1 min | 0.02 s |

[a] Early stopping: measurements until best validation loss.

In Table 16, the number of epochs and the total training time for the data-driven approaches are shown. For Step 1, both the UNet and DDUNet need many epochs and comparable training time to converge; however, for the third step and the full pipeline, the DDUNet significantly reduces both the number of epochs and the total training time required to reach convergence.

## A.4 HARDWARE SPECIFICATIONS

The $2\times2$ DDU-Net$_{101dp}$ and UNet$_{101dp}$ models, trained on the large data-driven dataset of 101 samples, were trained and evaluated on a server using NVIDIA V100 GPUs with 32 GB memory. All training was conducted using PyTorch 2.1.0 with CUDA 11.6 acceleration.

Training and evaluation of the LGCNN model were performed on a single NVIDIA A100-SXM4 GPU. Data generation was carried out on a dual-socket system equipped with AMD EPYC 9274F CPUs.

Table 16: Training measurements for the data-driven approaches trained on the *101dp* dataset: UNet$_{101dp}$ and 2×2 DDUNet$_{101dp}$.

|  | Epochs[a] | Training Time[a] |
|---|---|---|
| **1st Step** | | |
| UNet$_{101dp}$ | 738 | 5.497 hours |
| 2×2 DDUNet$_{101dp}$ | 735 | 4.787 hours |
| **3rd Step** | | |
| UNet$_{101dp}$ | 726 | 8.343 hours |
| 2×2 DDUNet$_{101dp}$ | 303 | 3.508 hours |
| **Full Pipeline** | | |
| UNet$_{101dp}$ | 267 | 3.680 hours |
| 2×2 DDUNet$_{101dp}$ | 97 | 1.222 hours |

[a]Early stopping: measurements until best validation loss.

### A.5 GLOSSARY

A list of the most relevant physical properties used in our paper is provided in Table 17.

Table 17: Glossary of Abbreviations.

| Abbr. | Parameter |
|---|---|
| $t$ | time |
| $X(t_0)$ | property X at initial time |
| $X(t_{\text{end}})$ | property X at quasi steady-state |
| $X_{pred}$ | predicted property X |
| $i$ | positions of heat pumps |
| $Q_{inj}$ | injected mass rate |
| $\Delta T_{inj}$ | injected temperature difference |
| $k$ | permeability |
| $p$ | hydraulic pressure |
| $g = \nabla p$ | hydraulic pressure gradient |
| $\vec{v} = (v_x, v_y)$ | flow velocity |
| $\vec{s}$ | both streamline fields |
| $s$ | central streamlines after all $i$ |
| $s_o$ | streamlines with transversal offset to $i$ |
| $T$ | temperature |

## Supplementary Material

The supplementary material contains the raw datasets, the most important trained models and the code basis for preparing the raw data to train on, separate training routines and evaluation protocols for LGCNN (on real or synthetic/random permeability fields), $UNet_{3dp}$, experiments with *3dp*; and on the other hand everything with *101dp*: $DDUNet_{101dp}$, $UNet_{101dp}$, experiments with *101dp*.

Raw datasets:

- Dataset of random permeability with 3+1 datapoints (*3dp* + 1dp)
- Dataset of random permeability with 101 datapoints (*101dp*)
- Dataset of real permeability with 4+1 datapoints

Trained models (including hyperparameters):

- vanilla approaches trained on random permeability fields, *3dp*: $UNet_{3dp}$
- vanilla approaches trained on random permeability fields, *101dp*: $DDUNet_{101dp}$, $UNet_{101dp}$
- LGCNN on random permeability fields, *3dp*
- LGCNN experiment: replace isolated steps 1 and 3 with $DDUNet_{101dp}$, $UNet_{101dp}$
- LGCNN on real permeability fields, *4dp*

Code (including training and evaluation routines):

- Repository of 101dp-vanilla approaches ($DDUNet_{101dp}$, $UNet_{101dp}$, also experiment "replace isolated steps")
- Repository LGCNN + $UNet_{3dp}$ (including preparation script for datasets to prepare datasets, for all models and approaches)

All supplementary material can be accessed via SURFDrive, through this link: https://surfdrive.surf.nl/files/index.php/s/f3Oqg3ufir9T9LL.

