# OpenReview forum: "Resolving Extreme Data Scarcity by Explicit Physics Integration: An Application to Groundwater Heat Transport"
_ICLR.cc/2026/Conference — ICLR 2026 Conference Withdrawn Submission_

### Official Review · Reviewer_jMn3 · 2025-10-24

**Soundness:** 2
**Presentation:** 3
**Contribution:** 1
**Rating:** 2
**Confidence:** 4

**Summary:**

This paper proposes LGCNN, a hybrid approach for predicting subsurface temperature fields from groundwater heat pumps. The key idea is to split the problem into three steps: use CNNs to predict velocity from pressure/permeability (local physics), compute streamlines numerically (global transport), then use another CNN to predict temperatures. They test on synthetic fields and some real Munich data, claiming this reduces data requirements and scales to larger domains.

**Strengths:**

The operator splitting concept is sensible - recognising that CNNs struggle with long-range advective transport and handling it with a numerical surrogate is the right intuition. The physics decomposition is clean.

**Weaknesses:**

1. Training on 3 datapoints (literally 1 train, 1 val, 1 test) and claiming "strongly reduced data requirements" doesn't pass the smell test. Even with the 101-sample comparison, there's no proper cross-validation or statistical validation.

2. You cite this work, Pelzer et al. (2024), which describes essentially the same two-stage idea (numerical surrogate for global transport + CNN for local processes). What exactly is new here beyond that paper? The three-step breakdown feels like an implementation detail rather than a conceptual advance.

**Questions:**

- Could you expand on the novelty of your work in relation to Pelzer et al?

---

### Official Review · Reviewer_T6Sk · 2025-10-31

**Soundness:** 3
**Presentation:** 1
**Contribution:** 1
**Rating:** 2
**Confidence:** 4

**Summary:**

This paper proposes Local-Global CNN (LGCNN), which is a deep learning model for subsurface temperature and velocity fields for Ground Water Heat Pumps. LGCNN combines CNNs and a numerical solver for the initial value problem to leverage the advantages of both. More specifically, a CNN (U-Net) predicts velocity values from local relations, the numeric solver computes the streamlines, and a second CNN computes the temperatures. The proposed method is compared against two U-Net variations (purely data-driven), including one tailored for large domains (DDUNet). The datasets applied in the evaluation are based on real and synthetic permeability fields and include synthetic smaller and larger scale datasets and a larger dataset with measurements from Munich. Results using multiple metrics (MAE, MSE, Maximum Absolute Error, etc.) show that the proposed model can scale to large datasets and achieves lower error rates than the data-driven alternatives.

**Strengths:**

- The paper is motivated by a real and relevant application

- The idea of combining ML and classical methods to speed up simulations is promising

- The experiments are based on both real and synthetic datasets

**Weaknesses:**

- The contributions of the paper to ML are not clear: the paper focuses on a narrow application and proposes a simple solution integrating CNNs and a numerical solver. It is not clear how the proposed approach will lead to advances in ML for simulations. Moreover, the paper doesn’t contextualize the work within the large related literature on simulations recently published at ICLR, NeurIPS, ICML, AAAI, IJCAI, etc. A quick look at the citations shows that the paper is much more focused on the application domain, which is pretty narrow compared to other problems that have attracted the interest of the ML community, such as weather forecasting.


- The writing of the paper can be greatly improved: the paper is not well-written. The experiments are divided into two different sections, and the reader has to move back and forth to compare the results. The captions of the figures are not helpful in describing the take-home message from the results. Moreover, the problem could be made more abstract for an ML audience, since most readers will not have expertise on groundwater heat transport. Details about the application domain could be moved to the experiments, and maybe a motivation section. Potential impact on other related applications should be discussed in more detail.

- The baselines considered in the paper seem weak: the baselines considered in the paper are just U-Nets. These might be the approaches considered by practitioners, but in the ML community, there are many alternatives that are well-known, such as Neural Operators, Graph Neural Networks, Physics-Informed Neural Networks, Universal Differential Equations, etc. Some of these approaches were only discussed in text, but that is not sufficient.

**Questions:**

1) What are the fundamental contributions of this paper to ML beyond the specific application considered?

2) Why other ML approaches for physics-based simulations (see Weaknesses) have not been considered in the experiments?

---

### Official Review · Reviewer_Nyb5 · 2025-11-01

**Soundness:** 2
**Presentation:** 3
**Contribution:** 2
**Rating:** 4
**Confidence:** 2

**Summary:**

The authors propose LGCNN, a physics-inspired CNN hybrid architecture for the estimation of groundwater heat transport. It seems to be a non-trivial application due to a multitude of environmental factors and long-range dependencies. Evaluations show LGCNN out-performs other tested methods on the chosen dataset, both on training and test set. Authors claim their method also scales to larger regions: analysis is done by training on smaller regions of the dataset and validating on larger ones.

**Strengths:**

- Results sound promising: the model seems to learn and generalize well from few training samples.
- Models and data will be available after publication for reference.
- The architecture and general idea of physical formulas inside/between CNN seem interesting

**Weaknesses:**

- Comparisons are lacking: as far as I can tell the proposed LGCNN is never compared directly to other state of the art solutions or even non-deep learning solutions, this makes it difficult to judge how useful LGCNN is in practice.
- Also, the novelty is not entirely clear: physics informed neural networks are known and applied to a variety of domains, what makes LGCNN special in this regard? Could LGCNN be compared to other, established, physics informed neural network architectures? Comparisons seem to focus on variants of LGCNN with different backbones and training hyperparameters but not different architectures.
- Figures are not explained well. For example, Figure 1: what are I, P, T, s and s_o? There are some formulas in the caption, but these should also be explained in more detail in the text as this is the main contribution/idea of the paper and should thus be clear and understandable even for people outside the domain.
- The visual assessment might make sense, but it should also be explained better what is shown, for example Figure 2: the LGCNN seems to have a white border around the data, is this an effect of the model or the presentation? Also, the input data and result should be explained more.

**Questions:**

- What is the size of the model?
- The dataset seems relatively small, if the model is large, how is overfitting addressed? Just via weight decay? It’s not addressed in the main part of the paper at all.

---

### Official Review · Reviewer_zSSc · 2025-11-04

**Soundness:** 3
**Presentation:** 3
**Contribution:** 2
**Rating:** 4
**Confidence:** 4

**Summary:**

This paper proposes a Local–Global CNN (LGCNN) framework to model advection-dominated subsurface heat transport under extreme data scarcity. The method decomposes the problem into: (1) a CNN that predicts steady-state flow fields, (2) a numerical solver that integrates streamlines to capture global advection, and (3) another CNN that predicts temperature fields conditioned on local features and streamline embeddings. This hybrid design leverages physical priors to achieve strong accuracy with as few as one or two training simulations and scales to larger spatial domains without retraining. Experiments on synthetic and real groundwater data show that LGCNN matches or surpasses standard UNet-based surrogates trained with significantly more data.

**Strengths:**

1) Well-motivated physics factorization. The advection-dominated regime (Pe ≫ 1) motivates separating global transport from local effects, with streamlines serving as an informative, compact conditioning signal. This reduces the burden on CNN receptive fields and training data.

2) Data efficiency under scarcity. Matching UNet/DDUNet trained on ~73–101 datapoints using only 1–3 simulations for LGCNN in critical steps is impressive and practically important for scientific settings with expensive labels.

3) Robust scaling to larger domains. The method scales to 4× larger domains (synthetic) and extended domains (real Munich maps) without retraining, owing to the global step handling long-range transport and patch-based training for local steps.

4) Careful ablations. The paper tests training sequence (simulated vs predicted v for Step 3), zero-padding, data partitioning, and the role of central vs offset streamlines (value fading vs binary masks), giving insight into what makes the pipeline work.

5) Real-world transfer. Demonstrates applicability on measured permeability maps with many interacting heat sources—beyond pairwise/isolated pump scenarios common in prior work.

**Weaknesses:**

1) Pipeline error propagation & calibration. The full pipeline’s error notably increases vs. Step-3-isolated results (using simulated v). There’s limited analysis of where velocity errors matter most (e.g., bifurcations), how sensitive temperature predictions are to streamline integration tolerances, or whether outputs are calibrated (e.g., reliability vs PAT/SSIM). A quantitative uncertainty or sensitivity analysis is missing.

2) Global surrogate design choices. The IVP solver and 2D raster embedding (with linear fading) are somewhat ad-hoc; accuracy vs. step size/tolerance, integrator choice (Radau vs RK4), and the width/number of offset streamlines are not systematically optimized or analyzed across regimes.

3) Comparative breadth. While baselines include UNet, DDUNet, and brief notes on PINNs/FNOs, there’s no head-to-head with global receptive field architectures like Swin-UNet variants, dilated/atrous stacks, or modern neural operator variants designed for multi-source fields (e.g., localized operator kernels) under equal data budgets; the FNO discussion is qualitative and hardware-limited.

4) Physical scope limitations. The method is steady-state, 2D, and one-way coupled (no thermal feedback on flow). The authors acknowledge this, but it limits immediate deployment where transient or 3D effects and thermo-hydraulic coupling matter. Benchmarks on even coarse 3D or short-horizon transients would strengthen claims of generality.

5) Metrics vs application risk. PAT at 0.1°C and SSIM are helpful, but siting/interaction decisions could depend on tail behavior (e.g., maximum plume extents, constraint violations). More task-level metrics (e.g., false-negative plume overlap beyond regulation thresholds) and cost/benefit versus full simulators would improve practical relevance.

**Questions:**

1) Uncertainty & sensitivity. Can you provide calibrated uncertainty for T (e.g., via ensemble of streamline perturbations or dropout), and a sensitivity study for (a) streamline integrator step/tolerance, (b) number and offset of auxiliary streamlines, and (c) pressure-gradient mis-specification? This would contextualize PAT under real deployment noise.

2) Where do velocity errors hurt? Please quantify how local v errors map to downstream temperature errors—e.g., plot error vs. distance along principal streamlines; analyze bifurcation regions where small k variations flip paths. Could a corrective local refinement near bifurcations reduce the pipeline gap?

3) Alternative global modules. Did you try (or could you include) comparisons with (i) dilated CNN backbones, (ii) attention-augmented UNets, or (iii) localized neural-operator kernels trained under the same 3-datapoint budget? Even a scaled-down study would position LGCNN more broadly.

---

### Note · Authors · 2025-11-27

I have read and agree with the venue's withdrawal policy on behalf of myself and my co-authors.